# Target-Aware Bandit Allocation for
# Scalable Surrogate Optimization in Chemical Space

Mohammad Haddadnia [1 2 3]  Yuvan Chali [1 2 4]  Abhilash Jayaraj [1 2]  Constance Kraay [5]  Joana Reis [1 2]
Felix Strieth-Kalthoff [6 7]  Haribabu Arthanari [1 2]

## Abstract

Identifying high-utility candidates from massive discrete spaces under expensive evaluations is a recurring challenge across the sciences, with structure-based drug discovery as a prominent example. While surrogate-based optimization can increase sample efficiency by reducing the number of expensive evaluations, modern molecular libraries have reached billions to trillions of compounds, making full-library surrogate inference itself a major computational bottleneck. We introduce BOBA, a bandit-guided surrogate optimization framework that eliminates full-library inference by adaptively allocating computation across partitions of the action space. By treating partitions as arms in a multi-armed bandit, BOBA concentrates inference and evaluations on empirically promising partitions while maintaining principled exploration. Experiments on real-world synthesis-on-demand libraries demonstrate that optimism-under-uncertainty bandits, combined with meaningful action space partitioning, are essential for effective allocation of inference and evaluations. Our findings reveal a tunable tradeoff between screening performance and surrogate inference cost, which supports practical optimization over current libraries, and establishes a viable route to ultra-large library virtual screening.

[1]Dana-Farber Cancer Institute, Boston, MA, USA [2]Department of Biological Chemistry and Molecular Pharmacology, Harvard Medical School, Boston, MA, USA [3]Department of Biostatistics, Harvard T.H. Chan School of Public Health, Boston, MA, USA [4]Department of Chemistry, Yale University, New Haven, CT, USA [5]Graduate Program in Biophysics, Harvard University, Cambridge, MA, USA [6]University of Wuppertal, School of Mathematics and Natural Sciences, Wuppertal, Germany [7]University of Wuppertal, Interdisciplinary Center of Machine Learning and Data Analytics, Wuppertal, Germany. Correspondence to: Felix Strieth-Kalthoff <strieth-kalthoff@uni-wuppertal.de>, Haribabu Arthanari <hari_arthanari@hms.harvard.edu>.

*Proceedings of the 43$^{rd}$ International Conference on Machine Learning*, Seoul, South Korea. PMLR 306, 2026. Copyright 2026 by the author(s).

## 1. Introduction

The earliest stages of drug discovery are dominated by a large and resource-intensive search problem: finding molecules that potently and selectively engage a biological target, and can be advanced into therapeutic leads. This hit identification stage remains a major bottleneck in the drug discovery pipeline, both in terms of cost and time. Computational approaches offer a fast and inexpensive route to exploring diverse sets of candidates, and recent advances in algorithms, computing, and ultra-large compound libraries have renewed optimism about their impact (Lyu et al., 2019; Gorgulla et al., 2020; Grygorenko et al., 2020; Sadybekov et al., 2021; Eisenhuth et al., 2025). Importantly, costs increase sharply at each downstream stage of the drug discovery pipeline (Morgan et al., 2011), which makes the quality of early computational decisions disproportionately critical.

Computational candidate selection in drug discovery is fundamentally a discrete, sequential decision-making problem. Although the space of possible molecules is astronomically large, with estimates ranging from $10^{60}$–$10^{200}$ (Restrepo, 2022), only a small fraction can be practically synthesized from commercially available building blocks at reasonable cost (Papidocha et al., 2026). In silico drug discovery therefore operates over large but finite libraries of synthetically accessible compounds (Shoichet, 2004), framing hit identification as a search over a large but discrete action space under budget constraints.

Active Learning (Settles, 2012) and Bayesian Optimization (Garnett et al., 2012; 2015; Jiang et al., 2018; Garnett, 2023) have become central tools for navigating these settings. While libraries have historically been evaluated exhaustively using molecular docking, a physics-informed simulation that estimates binding affinity between a molecule and a biological target, this strategy became prohibitively expensive as library sizes grew to billions of molecules. At the same time, recent studies have underscored the importance of screening larger and more structurally diverse libraries to increase true-hit rates and potency (Liu et al., 2025; Lyu et al., 2023; Gloriam, 2019). This need motivated surrogate-based optimization techniques that evaluate

only a small subset of candidates and use learned models to guide subsequent selection (Reker & Schneider, 2015; Pyzer-Knapp, 2018; Reker, 2019; Graff et al., 2021).

Recent years have witnessed yet another regime shift. Advances in make-on-demand chemistry have expanded accessible libraries from billions to trillions of molecules (Enamine Ltd.; Hoffmann & Gastreich, 2019; Warr et al., 2022; Gorgulla et al., 2023), altering the computational cost structure. At these scales, cost is no longer dominated solely by expensive physics-informed evaluations. Instead, inference over the candidate set itself becomes a bottleneck: even a single forward pass of a surrogate model over the full library can be prohibitively expensive. This bottleneck violates a core assumption of standard active learning pipelines that surrogate inference is negligible relative to evaluation cost (Frazier, 2018; Garnett, 2023).

This shift in the cost hierarchy reframes molecular discovery as a large-scale discrete decision-making problem under dual constraints on evaluation and inference. To address this challenge, we introduce BOBA (Bayesian Optimization with BAndits), which explicitly accounts for inference cost by combining structure-aware partitioning of the action space, bandit-based allocation across partitions, and surrogate-guided optimization within partitions. By decoupling global and local search, BOBA enables efficient candidate selection without exhaustive inference over the action space. Systematic benchmarks on real-world drug discovery data demonstrate the critical importance of a) bandit strategies which explicitly account for uncertainty, rather than relying randomized exploration, and b) a chemically sensible partitioning of action space. Our empirical results demonstrate a tunable tradeoff between optimization performance and inference cost, which allows the optimization efficiency of complete-inference BO to be largely retained, at substantially reduced inference cost. Scaling experiments up to approximately $10^8$ molecules show that this tradeoff becomes increasingly favorable as library size grows, and a simple theoretical analysis identifies the number of partitions as the key control parameter governing the balance between inference savings and bandit regret. These results lay the foundation for scaling to virtual libraries that contain billions to trillions of candidates.

## 2. Preliminaries

This section introduces the optimization setting considered in this work and reviews relevant concepts from Bayesian optimization and multi-armed bandits (MAB).

### 2.1. Virtual Screening as Large-Scale Discrete Optimization

We consider the problem of computational candidate selection in virtual screening. Let $\mathcal{X} = \{x_1, \ldots, x_N\}$ denote a finite library of candidate molecules, where $N$ may range from millions to trillions. Each molecule $x \in \mathcal{X}$ is associated with an unknown property of interest $f(x) \in \mathbb{R}$, such as docking score, binding affinity, or experimental activity, which can only be accessed through an expensive evaluation (e.g., docking or wet-lab assay).

The goal of virtual screening is to efficiently identify molecules with high values of $f(x)$ using as few evaluations as possible, framing virtual screening as a discrete optimization problem,

$$x^* = \arg\max_{x \in \mathcal{X}} f(x),$$

under a limited evaluation budget $T \ll |\mathcal{X}|$.

In addition to evaluation cost, we explicitly consider the computational cost of surrogate inference over $\mathcal{X}$. In modern make-on-demand settings, $|\mathcal{X}|$ is sufficiently large that exhaustive scoring of all candidates using a learned model is itself infeasible. We therefore distinguish between: (i) *evaluation cost*, incurred when querying $f(x)$, and (ii) *inference cost*, incurred when computing surrogate predictions over subsets of $\mathcal{X}$. This distinction is central to the problem setting addressed in this work.

### 2.2. Surrogate Modeling and Bayesian Optimization

Bayesian optimization (BO) addresses black-box optimization by maintaining a probabilistic surrogate model over $f$ (Garnett, 2023). Given a dataset $\mathcal{D}_t = \{(x_i, y_i)\}_{i=1}^{t}$ with $y_i = f(x_i) + \epsilon_i$, the surrogate defines a posterior predictive distribution

$$p(f(x) \mid \mathcal{D}_t),$$

which is used to construct an acquisition function $a_t(x)$ that balances exploration and exploitation.

In discrete settings, BO typically proceeds by selecting

$$x_{t+1} = \arg\max_{x \in \mathcal{X}} a_t(x).$$

Unless $|\mathcal{X}|$ is large, this step is commonly approximated by exhaustively evaluating $a_t(x)$ over $\mathcal{X}$. However, this assumption becomes invalid for ultra-large libraries, where surrogate inference over $\mathcal{X}$ constitutes a computational bottleneck. This computational barrier motivates algorithmic strategies that avoid full-library surrogate evaluations while retaining the benefits of Bayesian decision-making.

### 2.3. Multi-Armed Bandits

Multi-armed bandits (MAB) (Robbins, 1952; Lattimore & Szepesvári, 2020) formalize sequential decision-making un-

der uncertainty when limited resources must be allocated among competing alternatives. At each round $t$, an agent selects an arm $k \in \{1, \ldots, K\}$ and observes a stochastic reward drawn from an unknown distribution associated with that arm. Bandit algorithms adaptively trade off exploration and exploitation to identify high-reward arms or to maximize cumulative reward.

In this work, each arm corresponds to a subspace of chemical space, and the observed reward summarizes the utility of recent evaluations from that region. For the following discussion of Bandit policies, let $\hat{\mu}_k$ be the empirical mean reward of arm $k$, and $n_k$ the number of times arm $k$ has been selected.

$\epsilon$**-Greedy** selects the empirically best arm with probability $1 - \epsilon$, and explores by selecting an arm uniformly at random with probability $\epsilon$. Although simple and computationally inexpensive, $\epsilon$-greedy does not explicitly account for uncertainty, which can lead to inefficient allocation when the number of arms is large.

**Softmax Sampling** selects arms stochastically in proportion to their empirical mean rewards,

$$P(k_t = k) \propto \exp(\tau \hat{\mu}_{k,\, t-1}),$$

where $\tau > 0$ is an inverse-temperature parameter controlling how strongly the distribution concentrates on high-reward arms. Softmax provides a smooth tradeoff between exploration and exploitation, interpolating between uniform sampling and greedy selection.

**Upper Confidence Bound (UCB1)** selects arms optimistically based on both empirical performance and uncertainty (Auer et al., 2002). At round $t$, UCB1 chooses the arm maximizing

$$\hat{\mu}_{k,\, t-1} + c\sqrt{\frac{2 \log t}{n_{k,\, t-1}}},$$

where $c > 0$ controls the exploration–exploitation tradeoff.

Formal definitions and implementation details are provided in Appendix A.3.

## 2.4. Partitioning Methods for Chemical Space

Let $\mathcal{X} = \{x_1, \ldots, x_N\}$ denote a virtual library of molecules $x_i$. When partitioning such virtual libraries, we consider dividing $\mathcal{X}$ into $K$ disjoint subsets $\{\mathcal{X}_1, \ldots, \mathcal{X}_K\}$. Typically, such partitioning is performed in a molecular feature space induced by a representation $\phi(x) \in \mathbb{R}^d$, which can be either learned or engineered.

### 2.4.1. MOLECULAR FEATURES

**Topological features** encode molecular structure as graph-derived patterns capturing atom connectivity and substructures (e.g. paths, cycles, and local neighborhoods). They often provide sparse, discrete representations ("fingerprints") optimized for similarity search.

**Physicochemical descriptors** are engineered low-dimensional features that summarize global molecular properties derived from the graph structure (e.g. molecular weight, hydrogen bond donor count, polarity, or solubility). They offer interpretable, physically meaningful signals, but may miss fine-grained structural detail. The full list of descriptors used in this work is provided in Appendix B.

**Foundation model embeddings** are dense, learned representations produced by deep neural networks pretrained on large molecular corpora using self-supervised objectives. Pre-trained networks can include graph neural networks and language models operating on SMILES, a string-based encoding of the molecular graph structure. In this work, we focus on embeddings from the T5Chem model (Christofidellis et al., 2023), a chemistry-specific variant of the T5 architecture (Raffel et al., 2020) pretrained on large corpora of molecular structures and textual descriptions. Prior work has shown that these embeddings capture rich chemical diversity and improve performance in molecular property prediction and active learning compared to traditional fingerprints (Kristiadi et al., 2024).

### 2.4.2. PARTITIONING TECHNIQUES

**Feature-based stratification** deterministically partitions molecules using fixed intervals along a small number of hand-crafted features, yielding axis-aligned regions in feature space. This approach has previously been used to partition ultra-large chemical libraries into chemically coherent regions for virtual screening (Gorgulla et al., 2023).

$k$**-Means Clustering** groups data by assigning points to the nearest of $k$ learned centroids in feature space, optimized by minimizing within-cluster variance.

$$\sum_{k=1}^{K} \sum_{x \in \mathcal{X}_k} \|\phi(x) - \mu_k\|_2^2,$$

As a baseline, we construct unstructured subspaces by randomly permuting $\mathcal{X}$ and dividing it into $K$ equally sized bins. This randomization removes all chemical structure while preserving cluster sizes, isolating the effect of meaningful space decomposition.

# 3. Method

Building on the preliminaries, we formalize the problem of surrogate-based discovery from ultra-large molecular libraries, and introduce the algorithmic framework of BOBA. We also summarize related works relevant to the method proposed herein.

## 3.1. Problem Setting

Given a large discrete library $\mathcal{X}$, an unknown black-box function $f$ (e.g. a docking score), and and a budget $T$ on the number of function evaluations, we seek an algorithm that sequentially selects candidates $\mathcal{C} = \{x_1, \ldots, x_T\} \subset \mathcal{X}$ under these constraints. At each time point $t$, we assume access to a molecular representation $\phi(x) \in \mathbb{R}^d$, and a parametric or nonparametric surrogate model trained on all previous observations $\mathcal{D}_t = \{(x_i, f(x_i))\}_{i=1}^t$. Let $\mathcal{T}_m \subset \mathcal{X}$ denote the set of elements corresponding to the largest values of $f$.

In this scenario, a design strategy should select candidates such that

  a) the number of top-$m$ elements recovered (i.e., $|\mathcal{C} \cap \mathcal{T}_m|$) is maximized.

  b) the number of surrogate inferences is minimized.

## 3.2. Formulation of BOBA

BOBA integrates surrogate-based optimization with structured allocation of inference and evaluation budgets. The fundamental concept entails decomposing $\mathcal{X}$ into $K$ subsets $\{\mathcal{X}_1, \ldots, \mathcal{X}_K\}$, each of which serves as an arm in a multi-armed bandit problem.

At each iteration, a bandit algorithm selects a subset $\mathcal{X}_k$ based on past observations. Pulling arm $k$ triggers a localized optimization procedure within $\mathcal{X}_k$: surrogate inference is performed only on this subset, an acquisition function is evaluated locally, and a small batch of $B$ candidates is selected for expensive evaluation. The newly observed data are then added to $\mathcal{D}_t$, and the surrogate is updated. Ultimately, the observed reward becomes an aggregate metric of the utility of recent evaluations from that subspace (e.g., average docking score) and is used to guide future allocation (see Algorithm 1).

This perspective reframes large-library screening as a hierarchical decision problem: rather than repeatedly scoring all candidates, the algorithm first decides *where* in chemical space to invest computational resources, and only then performs fine-grained surrogate-based selection within the chosen region. When subspaces differ in their intrinsic concentration of high-quality molecules, bandit algorithms provide a principled mechanism for identifying and exploit-

---

**Algorithm 1** BOBA: Bayesian Optimization with Bandits over Clustered Subspaces

---

**Require:** Library $\mathcal{X}$; expensive oracle $f(\cdot)$; featurizer $\phi(\cdot)$; number of subspaces $K$; bandit algorithm $\mathcal{B}$; surrogate model class $\mathcal{M}$; acquisition function $a(\cdot)$; rounds $T$; batch size $B$

 1: **Cluster** $\mathcal{X}$ into $K$ disjoint subspaces $\{\mathcal{X}_1, \ldots, \mathcal{X}_K\}$ using features $\phi(x)$
 2: Initialize dataset $\mathcal{D}_0 \leftarrow \emptyset$ (or a small seed set)
 3: Initialize bandit state for arms $k \in \{1, \ldots, K\}$ in $\mathcal{B}$
 4: **for** $t = 1, \ldots, T$ **do**
 5:     Select an arm (subspace) $k_t \leftarrow \mathcal{B}(\text{history})$
 6:     Fit/update a **global** surrogate $M_t \in \mathcal{M}$ on $\mathcal{D}_{t-1}$
 7:     **Localized inference:** compute acquisition scores $\{a_t(x)\}_{x \in \mathcal{X}_{k_t}}$ using $M_t$
 8:     Select a batch:
        $\mathcal{Q}_t \leftarrow \text{Top-}B$ elements of $\mathcal{X}_{k_t}$ by $a_t(x)$
 9:     Evaluate oracle: obtain $y_t(x) \leftarrow f(x)$ for all $x \in \mathcal{Q}_t$
10:     Update dataset: $\mathcal{D}_t \leftarrow \mathcal{D}_{t-1} \cup \{(x, y_t(x)) : x \in \mathcal{Q}_t\}$
11:     Compute arm reward (batch average):
        $r_t \leftarrow \frac{1}{B} \sum_{x \in \mathcal{Q}_t} y_t(x)$
12:     Update bandit state: $\mathcal{B} \leftarrow \text{UPDATE}(\mathcal{B}, k_t, r_t)$
13: **end for**
14: **Output:** $\arg\max_{(x,y) \in \mathcal{D}_T}(y)$

---

ing promising regions while continuing to explore uncertain ones. This formulation is central to BOBA, enabling adaptive, data-driven control over inference and evaluation budgets without requiring exhaustive scans of the action space.

Within this work we perform extensive experimental benchmarks of BOBA: First, we evaluate different bandit algorithms and analyze their impact on optimization performance and sample efficiency. Second, we systematically study the tradeoff between optimization performance and computational cost by varying the number and size of partitions, highlighting how granularity affects inference cost. Third, we examine how different choices of featurization and partitioning strategy influence optimization performance, comparing structured clustering approaches to random partitions.

## 3.3. Related Work

Active learning strategies are widely used in large-library virtual screening and have been reviewed extensively elsewhere.(Reker & Schneider, 2015) Here, we focus on prior work that combines active learning with explicit segmentation of large molecular libraries. In general, virtual libraries are constructed by combinatorial enumeration from sets of synthetic precursors (*synthons*), and several approaches exploit this structure to improve search efficiency. A common

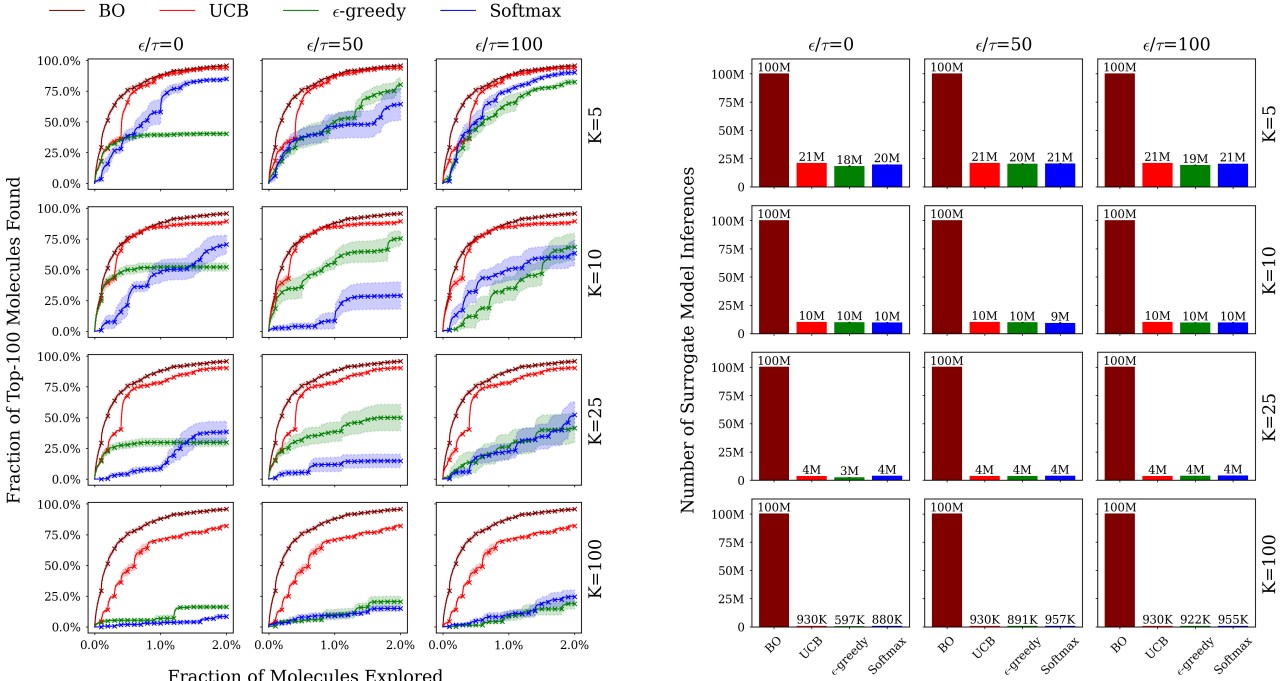

*Figure 1.* **Evaluation of Bandit algorithms, and observed tradeoff between performance and inference cost**. (**Left**) Optimization trajectories of BOBA with different Bandit algorithms as a function of the number of costly black-box evaluations. BO with full-library inference (maroon) is included as an upper boundary. Performance is quantified by the number of retrieved molecules from the top-100 candidates from the full library. (**Right**) Computational cost of the respective optimization runs, measured by the number of surrogate model inferences. All experiments are reported on the Enamine-5M library docked against CKB. Trajectories are shown as the mean over 5 independent runs from different seed populations. The shaded area indicates the standard error of the mean.

strategy is to perform search within this low-dimensional discrete synthon space, which avoids explicit enumeration of the full library. This setting has been addressed using active learning approaches (Grigg et al., 2025; Kozyrev et al., 2025), as well as multi-armed bandit formulations (Klarich et al., 2024; Zhao et al., 2025). Relatedly, the synthon-based structure of virtual libraries has also been exploited for hierarchical coarse-to-fine retrieval of candidate molecules (Nazarova et al., 2025).

Our work differs both in how structure is imposed on the search space and how this structure is exploited algorithmically. Existing approaches typically operate over a fixed synthon-defined hierarchical organization of the library. In contrast, BOBA uses explicit molecular-structure-aware partitioning, and combines it with bandit-based methods for dynamic allocation of sampling effort across partitions, and surrogate-based optimization for efficient candidate selection within each partition. This combination allows the algorithm to adaptively balance exploration and exploitation across heterogeneous regions of the library, rather than committing to a predefined partitioning or a single level of resolution. The resulting search strategy uses target feedback to allocate effort across learned molecular partitions,

without requiring the fixed synthon-level hierarchy assumed by fragment-based search methods.

## 4. Benchmarking Experiments

We benchmark BOBA on virtual libraries derived from ultra-large chemical spaces, as complete ground-truth docking scores are available for these libraries. Specifically, we randomly selected 5 million compounds from an enumerated library of approximately 69 billion molecules from ENAMINE REAL (Enamine Ltd.; Gorgulla et al., 2023), which we refer to as ENAMINE-5M. In addition, we sampled an independent set of 3.9 million compounds from ENAMINE's S-class small-molecule database (Enamine Ltd.), which we refer to as ENAMINE-S-3.9M. Both libraries were exhaustively docked against the protein targets NEDD4 (PDB: 9HT9; Maspero et al., 2025), and CKB (PDB: 3B6R; Bong et al., 2008). In addition, we use a virtual library of 2M molecules from ENAMINE's HTS database (ENAMINE-HTS), which was docked against a Thymidylate Kinase (TMK) by (Graff et al., 2021).

BOBA uses a feedforward neural network surrogate operating on fixed molecular features, with uncertainties approxi-

mated *via* a Linearized Laplace approximation (Daxberger et al., 2021) (see Appendix A.4 for more details). We additionally evaluate alternative uncertainty estimators, including MC dropout (Gal & Ghahramani, 2016) and SWAG (Maddox et al., 2019) (see Appendix A.5 and Figure 7); these experiments support the use of the Laplace approximation in all further experiments.

After a single-round initialization with randomly drawn samples from all partitions, BOBA follows the algorithm outlined in Algorithm 1. From a bandit-selected partition, all molecules are scored by the neural network surrogate, and a batch of $B$ candidates are selected using the Upper Confidence Bound as the acquisition function. Across all methods, we run $T = 20$ rounds of selection, with a batch size of $B = 5,000$ for a total of $TB = 100,000$ oracle evaluations. In the main text, we report optimization trajectories for the number of top-100 molecules from the full library that were found by the algorithm, which is motivated by budget constraints on downstream experimental validation. Trajectories for the top-1000 and top-10000 molecules are provided in the Appendix C, specifically Figures 8, 9, and 10.

### 4.1. Bandit Exploration Strongly Impacts Screening Performance

We first examine how the choice of bandit algorithm affects optimization performance. Specifically, we evaluate $\epsilon$-greedy, softmax sampling, and UCB1 bandits across different levels of exploration, and under variation of the granularity of library partitions. The resulting optimization trajectories and inference cost estimates on the Enamine-5M library, docked against CKB, are shown in Figure 1. As an upper-bound estimate of optimization performance, we compare against standard BO with full-library inference.

Empirically, we find that $\epsilon$-greedy and softmax exhibit substantial degradation of optimization performance as the number of library partitions increases. We attribute this behavior to the lack of principled uncertainty-guided exploration in these methods, which can lead to premature over-exploitation of suboptimal partitions, or insufficient exploitation of optimal ones. UCB1 instead assigns an optimism bonus to under-sampled arms, so uncertain partitions remain eligible for selection even when their empirical rewards are initially modest. Consistent with this mechanism, the per-arm selection frequencies in Appendix D, specifically Figure 11, show that UCB1 continues to allocate evaluations across multiple partitions throughout optimization, without collapsing onto a single early winner. Thus, UCB1 remains largely competitive with full-inference BO even at large numbers of partitions, recovering nearly the same number of top-100 ranked molecules at a substantially reduced inference cost. Notably, the trends remain

unchanged when considering the top-1000 and top-10000 candidate molecules from the full library (see Appendix C, specifically Figures 8, 9, and 10).

These results indicate that, particularly in regimes with many arms, optimism-based allocation is critical, which motivates the use of UCB1 in all subsequent experimental studies. Remarkably, even this relatively simple algorithm achieves strong performance, suggesting that more advanced uncertainty-aware bandit methods may offer additional benefits. However, their analysis is beyond the scope of this study.

Finally, even for the UCB1 algorithm, we observe a tradeoff between optimization performance and inference cost. This trade-off can be tuned by the user through the choice of partition granularity (that is, the number of clusters), depending on the requirements of a specific use case. The role of $K$ can be understood through a simple cost–regret decomposition. Partitioning reduces the cumulative surrogate inference cost from order $NT$ to approximately $NT/K$, but increasing $K$ also makes the bandit allocation problem harder because the algorithm must identify promising regions among more arms. Under a gap-free UCB-style regret term of order $\tilde{\mathcal{O}}(\sqrt{KT})$, minimizing a weighted objective of inference cost and allocation regret yields the heuristic scaling for choosing $K$ as $K^* \asymp N^{2/3}T^{1/3}$. This analysis, detailed in Appendix E, formalizes why larger libraries can support finer partitions while still requiring a sublinear choice of $K$.

### 4.2. Structured Subspaces are Critical

We next perform a systematic ablation to understand whether the gains of BOBA arise merely from partitioning the library, a persistent partitioning scheme, or from the inherent library structure captured by clustering.

#### 4.2.1. STATIC BOBA VS. DYNAMIC PARTITIONING

First, we compare BOBA to an unstructured baseline in which, at each iteration, standard BO is applied to a uniformly random subset of the library whose size matches the average cluster size in BOBA. Related subsampling strategies have been used in prior works to mitigate the cost of exhaustive surrogate inference (Wang-Henderson et al., 2023).

Figure 2 shows that BOBA substantially outperforms random subsampling across all settings. While both methods reduce inference cost to the same degree, subsampling repeatedly reallocates computation to arbitrary regions of chemical space. In contrast, BOBA defines persistent partitions and uses bandit feedback to adaptively concentrate evaluations in empirically promising regions. These findings indicate that BOBA achieves significant optimization performance gains by identifying persistent partitions of the

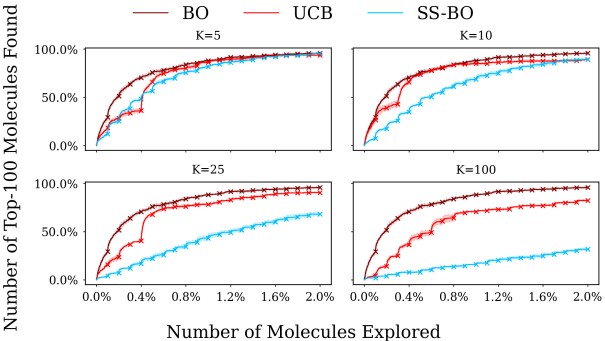

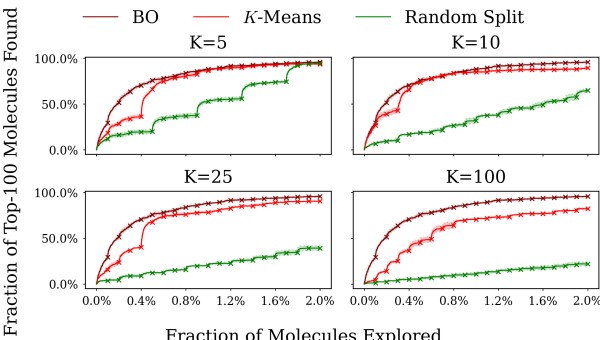

*Figure 2.* **Influence of static vs. dynamic partitioning.** Optimization trajectories of BOBA with UCB1 using $K$-Means-based partitions, compared to setting in which a partition for inference is randomly selected at each iteration. All experiments are reported on the Enamine-5M library docked against CKB. Trajectories are shown as the mean over 5 independent runs from different seed populations. The shaded area indicates the standard error of the mean.

*Figure 3.* **Influence of the partitioning scheme.** Optimization trajectories of BOBA with UCB1 using either $K$-Means-based partitions, or randomly assigned partitions. All experiments are reported on the Enamine-5M library docked against CKB. Trajectories are shown as the mean over 5 independent runs from different seed populations. The shaded area indicates the standard error of the mean.

action space, and adaptively allocating resources to these partitions.

### 4.2.2. STRUCTURED VS. RANDOMLY ASSIGNED PARTITIONS

Second, we investigate whether a static library partitioning alone is sufficient to maintain competitive optimization performance at reduced inference budget. Therefore, we compare BOBA using $K$-means-based molecular clusters to a variant in which molecules are randomly assigned once to $K$ equally sized, static subsets, which are then treated as arms by the same UCB1 algorithm. Both approaches use identical surrogates, acquisition functions, batch sizes, and bandit rules; they differ only in how the partitions are initially defined.

As shown in Figure 3, replacing the structured partitions obtained *via* $K$-means clustering with random partitions leads to a substantial degradation in optimization performance. Although both methods maintain persistent partitions and operate under identical inference budgets, only structured clustering yields arms that are systematically enriched for high-quality candidates. Analysis of the distribution of targets across partitions (see Appendix F, specifically Figures 12, 13, and 14, for further details) confirms the presence of significant inter-partition variance in the case of clustering, which is a prerequisite for effective exploration–exploitation across arms. These findings confirm that the effectiveness of BOBA depends critically on the chemical coherence of the partitions, rather than on a static partition of the library alone.

### 4.3. Effect of Feature Space

Having established the necessity of structured partitioning for achieving competitive optimization performance at reduced inference budget, we next evaluate how the molecular representation used to construct partitions impacts end-to-end BOBA performance. Therefore, we compare a feature space spanned by established physicochemical descriptors for drug discovery (Gorgulla et al., 2020) without further refinement, with the embedding space of a domain-specific language model (T5Chem).

Figure 4 demonstrates that, across all values of $K$, BOBA constructed in T5Chem embedding space consistently outperforms BOBA constructed in unrefined physicochemical descriptor space, recovering substantially more top-ranked molecules under identical budgets. These findings suggest that clustering in T5Chem embedding space produces partitions that are more closely aligned with the optimization objective, allowing the bandit to allocate more effectively across regions, while improving local surrogate generalization. In all experiments, the same molecular representation is used for both clustering and surrogate modeling. This design choice ensures that the geometry used to define bandit arms is consistent with the geometry used for local acquisition, avoiding discrepancies between global partitioning and local optimization. We therefore consider this the preferred default when the representation is sufficiently informative for the target task. At the same time, this design also implies that biases from a poorly aligned representation are propagated into both stages, motivating the partion-quality ablations discussed above. These conclusions align well with the findings by (Kristiadi et al., 2024), who reported that, even without task-specific fine-tuning, such embeddings provide an effective molecular representation across different tasks.

Overall, these results indicate that the effectiveness of BOBA is tightly coupled to the representational geometry of chemical space. While pre-trained molecular foundation models provide a robust starting point, we anticipate that integrating domain expertise into task-specific feature spaces can further improve or accelerate optimization with BOBA.

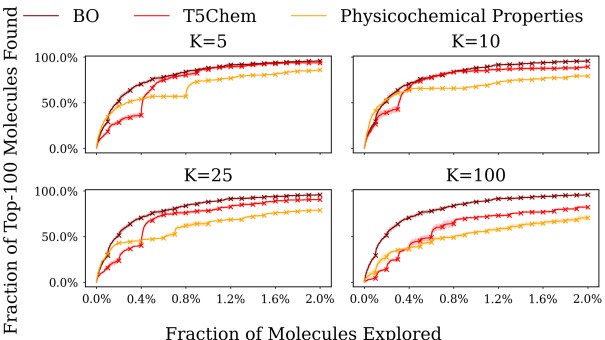

*Figure 4.* **Effect of feature space on end-to-end BOBA performance.** Optimization trajectories of BOBA, with partitions constructed using T5Chem language model embeddings, or using unrefined physicochemical descriptors. All experiments are reported on the Enamine-5M library docked against CKB. Trajectories are shown as the mean over 5 independent runs from different seed populations. The shaded area indicates the standard error of the mean.

### 4.4. Robustness Across Targets and Difficulty Regimes

Finally, we study the optimization behavior of BOBA with UCB1 on three additional tasks: the ENAMINE-HTS library docked against TMK (Graff et al., 2021), and the ENAMINE-S-3.9M library docked against CKB and NEDD4. The corresponding optimization trajectories, compared to BO with full-library inference as an upper-bound estimate, are shown in Figure 5.

Qualitatively, we find that the trends discussed in the previous sections are reproduced across tasks. As $K$ increases, we observe a similar sublinear decay of optimization performance, indicating a feasible tradeoff between optimization performance and inference cost. Notably, the absolute optimization scores on the ENAMINE-S-3.9M library are substantially lower than those on the ENAMINE-5M (CKB) benchmark, both for the upper-bound estimate and, by extension, for BOBA. This performance gap suggests that these tasks pose more challenging optimization problems. Indeed, analysis of the underlying library revealed a larger chemical diversity in the ENAMINE-S-3.9M library compared to the ENAMINE-5M library, which provides a possible explanation for the observed differences in absolute performance.

These results demonstrate that BOBA's performance–inference tradeoff and dependence on cluster granularity persist across targets and difficulty regimes, indicating that the framework generalizes beyond a single dataset or optimization landscape.

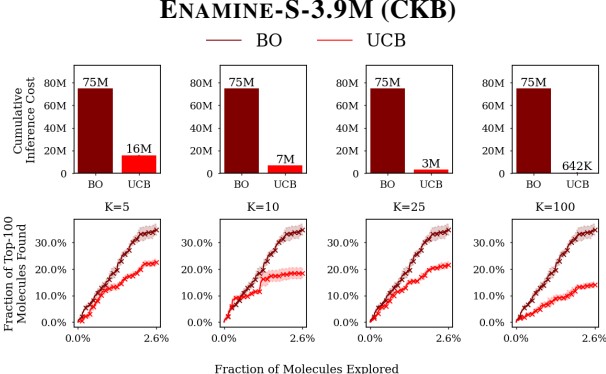

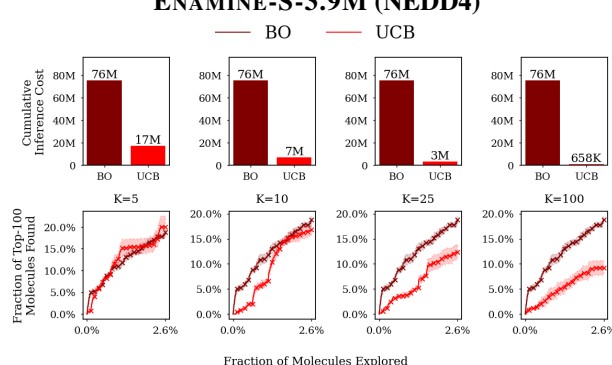

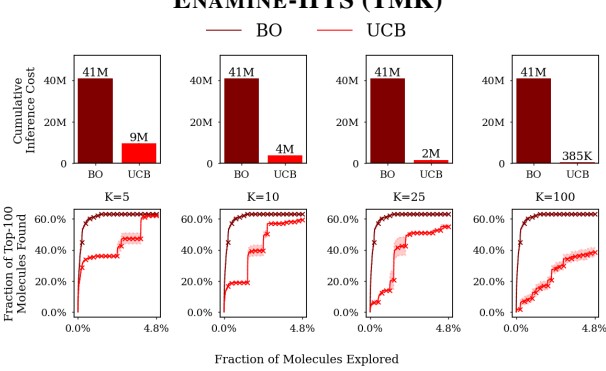

*Figure 5.* **BOBA across targets and difficulty regimes.** Optimization trajectories of BOBA on different optimization problems. Trajectories are shown as the mean over 5 independent runs from different seed populations. The shaded area indicates the standard error of the mean.

### 4.5. Scaling to Larger Libraries

The preceding experiments evaluate BOBA in settings where exhaustive surrogate inference is still possible, which enables direct comparison against full-library BO. To test whether the observed tradeoff remains relevant at larger

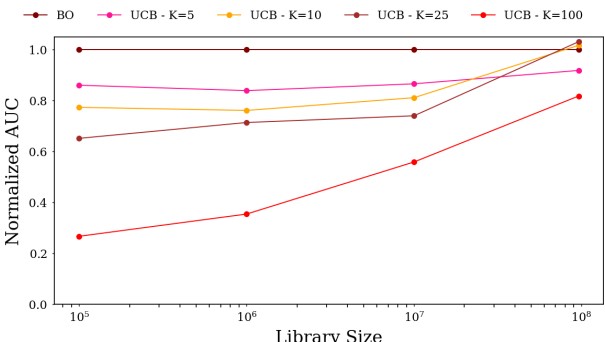

*Figure 6.* **Scalability of BOBA with library size.** Optimization performance of BOBA relative to full-library BO as the library size increases from approximately $10^5$ to $10^8$ molecules. Performance is measured as the area under the top-1000 retrieval curve and normalized by the AUC of the corresponding full-library BO run. All experiments use the ZINC library docked against AmpC, or subsets thereof.

scale, we further evaluate BOBA on the ZINC library docked against AmpC (Lyu et al., 2019) using nested libraries of approximately $10^5$, $10^6$, $10^7$, and $10^8$ molecules. For each library size, we compare the area under the top-1000 retrieval curve for BOBA against the corresponding full-library BO run, normalizing the BO performance to one.

As shown in Figure 6, BOBA's relative performance does not deteriorate as the candidate library grows. Instead, for several partition granularities, the normalized AUC approaches full-library BO at the largest tested scale. This trend is consistent with the core motivation of BOBA: as $N$ increases, full-library inference becomes increasingly costly, while localized inference over a selected partition remains controlled by the cluster granularity. The value of $K$ therefore acts as a user-facing knob that trades inference cost against the statistical difficulty of selecting among more arms. Appendix A.2 reports the corresponding wall-clock breakdown.

## 5. Conclusion and Outlook

Ultra-large make-on-demand libraries push surrogate-based optimization into a regime where surrogate inference and acquisition over the full candidate set can become as limiting as the expensive oracle itself. In this work, we introduce BOBA, a target-aware framework that avoids full-library inference by partitioning chemical space into persistent subspaces, and using a multi-armed bandit to allocate inference and evaluation to regions that empirically yield high-utility molecules.

Benchmark experiments highlight the critical influence of both the bandit exploration strategy and the structure of the partitioning scheme. Optimism-under-uncertainty bandits combined with clustering on foundation model em-

beddings consistently delivered robust performance across tasks, clearly outperforming clustering on unrefined physicochemical descriptors as well as unstructured baselines on randomized partitions. Looking forward, we anticipate that advances in bandit algorithms, particularly improved uncertainty quantification and rotting bandits to accommodate non-stationary partition rewards (Levine et al., 2017), will further enhance optimization performance. Coupling these methods with problem-specific, expert-refined molecular representations is a promising direction for further gains.

Our findings further reveal a clear trade-off between optimization performance and inference cost. Empirical results across increasingly large libraries indicate that the decay in optimization performance with increasing numbers of partitions is sublinear, making it feasible for practitioners to select the number of partitions, and therefore the inference cost, according to problem-specific constraints. This behavior is supported by a simplified cost-regret analysis, which identifies the number of clusters as a natural control parameter and predicts an optimal granularity that grows sublinearly with library size. While partitioning very large libraries introduces additional computational cost, the employed algorithm scales as $\mathcal{O}(nkid)$, dominated by the library size $n$, and constitutes a one-time processing expense that can be reused across targets. Assessing how broadly these results generalize across even larger libraries and more diverse targets remains the focus of ongoing work. Overall, this study represents a step toward principled decision-making at billion- to trillion-scale, in chemical discovery and other large discrete search problems, where both evaluation and inference are costly.

## 6. Limitations

The current implementation and empirical evaluation of BOBA are restricted to settings in which the virtual library is explicitly enumerable, and in which featurization and clustering of the full library remain computationally tractable. Conceptually, however, the proposed framework does not rely on explicit enumeration of candidate molecules. The combination of library partitioning, bandit-driven allocation of sampling effort across partitions, and surrogate modeling for candidate selection could instead be applied directly in synthon space. This extension would avoid the need for full library enumeration and global featurization. Instead, enumeration, featurization, and inference would be required only for global subsamples or small, tractable partitions. We leave the implementation and systematic benchmarking of this synthon-level formulation to future work.

## Acknowledgements

The authors thank Dr. Michael Emanuel for helpful discussions and Dr. Christoph Gorgulla for providing the Enamine-5M library. Y.C. acknowledges support from the BCMP Scholars Summer Undergraduate Research Program at Harvard Medical School. C.K. was supported by a Hertz Foundation Fellowship and previously by NIH/NIGMS Molecular Biophysics Training Grant T32 GM008313. F.S.-K. acknowledges funding from the Deutsche Forschungsgemeinschaft (DFG) under the Priority Program 2363 "Molecular Machine Learning" (grant no. 497260357). H.A. acknowledges support from Pivotal Life Sciences and NIH grant R35 GM158220 (NIGMS).

## Impact Statement

This work aims to accelerate the identification of promising candidates from ultra-large discrete libraries, motivated by applications in early-stage drug discovery. By improving the efficiency of computational candidate selection, such methods may reduce the cost and time required to identify therapeutic leads, with potential downstream benefits for global healthcare. The proposed framework addresses a general class of large-scale discrete optimization problems, and may therefore be applicable beyond drug discovery, e. g. in materials design, catalyst discovery, or related scientific settings. The societal consequences of such applications depend on the specific use case, but could be relevant to areas like energy storage and conversion, sustainable materials development, or improved resource and materials cycles.

At the same time, more efficient exploration of chemical space raises potential dual-use concerns. While the present approach is limited to computational prioritization and requires substantial expertise and experimental infrastructure to realize practical impact, responsible development and deployment are important considerations.

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

# A. Additional Implementation Details

## A.1. Fair Budgeting

Importantly, although clustering reduces the average search space per round from $N$ to approximately $N/K$, we do *not* reduce the batch size when using BOBA. One might consider evaluating only $B/K$ molecules per iteration within each subspace and increasing the number of rounds accordingly; however, this would equalize the total number of surrogate inferences between BOBA and standard BO, obscuring the regime we aim to study.

Instead, we fix both $T$ and $B$ across all methods. As a result, BOBA performs approximately $K$-fold fewer surrogate inferences than standard BO, directly exposing the tradeoff between inference cost and optimization performance.

## A.2. Runtime Breakdown

To complement the inference-count analysis in the main text, Table 1 reports wall-clock measurements for preprocessing and optimization on the ZINC–AmpC scaling benchmark. The measurements separate one-time library preprocessing costs, namely embedding computation and clustering, from the repeated optimization loop. Embedding computation dominates total runtime, but this cost is amortized across downstream targets because the same molecular representations and partitions can be reused. Within a fixed optimization campaign, the dominant avoidable cost is surrogate inference, which is precisely the component reduced by restricting acquisition evaluation to the selected partition.

| Dataset | Emb. | $K = 5$ | $K = 10$ | $K = 25$ | $K = 100$ | BO | BoBa-5 | BoBa-10 | BoBa-25 | BoBa-100 |
|---|---|---|---|---|---|---|---|---|---|---|
| 96M | 12774 | 541 | 649 | 810 | 855 | 2720 | 2137 | 2258 | 2318 | 2267 |
| 10M | 1281 | 88 | 89 | 96 | 99 | 469 | 361 | 286 | 321 | 319 |
| 1M | 129 | 3 | 12 | 14 | 18 | 57 | 50 | 41 | 37 | 39 |

*Table 1.* **Wall-clock runtime breakdown for the ZINC–AmpC scaling benchmark.** All values are reported in seconds. "Emb." denotes embedding computation, the columns labeled by $K$ denote clustering/tranching time for the corresponding number of partitions, and the final columns report optimization-loop runtime for full-library BO and BOBA. Model training and inference were performed on an NVIDIA GH200 Grace Hopper chip.

## A.3. Multi-Armed Bandit Algorithms

In this work, each clustered subspace $\mathcal{X}_k$ is treated as an arm in a multi-armed bandit (MAB) problem. At iteration $t$, selecting arm $k$ yields a scalar reward $r_{k,t}$, defined as the average docking score of the batch evaluated from that subspace. We maintain, for each arm $k$, the number of times it has been selected $n_k(t)$ and its empirical mean reward

$$\hat{\mu}_k(t) = \frac{1}{n_k(t)} \sum_{i=1}^{n_k(t)} r_{k,i}.$$

**Initialization.** All algorithms use a single-round initialization strategy: the initial batch is distributed across arms, with $B/K$ molecules seeded from each partition when $B$ is divisible by $K$. This seeding strategy initializes every arm without spending one full optimization round per arm, which is important when $T$ is small relative to $K$.

We consider the following bandit strategies.

### A.3.1. UCB1

Upper Confidence Bound (UCB1) selects the arm that maximizes an optimism-adjusted estimate of the mean reward (Auer et al., 2002):

$$k_t = \arg \max_{k \in \{1, \ldots, K\}} \left[ \hat{\mu}_k(t-1) + c\sqrt{\frac{2 \log t}{n_k(t-1)}} \right],$$

where $c > 0$ is a tunable exploration constant (set to $c = 1$ in all experiments unless otherwise stated). Arms are initialized by the single-round seeding strategy described above before the UCB1 rule is applied.

### A.3.2. $\epsilon$-GREEDY

The $\epsilon$-greedy strategy selects arms according to

$$k_t = \begin{cases} \arg\max_k \hat{\mu}_k(t-1), & \text{with probability } 1 - \epsilon, \\ \text{Uniform}(\{1, \ldots, K\}), & \text{with probability } \epsilon, \end{cases}$$

where $\epsilon \in [0, 1]$ controls the exploration rate. We report results for multiple values of $\epsilon$.

### A.3.3. SOFTMAX SAMPLING

Softmax (Boltzmann) exploration samples arms stochastically according to their empirical mean rewards:

$$P(k_t = k) = \frac{\exp(\tau \, \hat{\mu}_k(t-1))}{\sum_{j=1}^{K} \exp(\tau \, \hat{\mu}_j(t-1))}.$$

Here $\tau > 0$ is an inverse-temperature parameter controlling the sharpness of the distribution. Larger $\tau$ increasingly concentrates probability mass on the empirically best arm, while smaller $\tau$ approaches uniform sampling. Note that we parameterize softmax using $\exp(\tau x)$ rather than the more common $\exp(x/\tau)$; under this convention, larger $\tau$ corresponds to lower stochasticity.

## A.4. Bayesian Optimization

All screening methods in this work, including standard Bayesian optimization and BOBA, are built on a common surrogate modeling and acquisition framework. This shared framework ensures that observed differences arise from the allocation strategy rather than from differences in modeling capacity.

### A.4.1. SURROGATE MODEL

We model the unknown objective function $f(x)$ using Bayesian neural networks (BNNs) trained on the growing dataset $\mathcal{D}_t = \{(x_i, y_i)\}_{i=1}^{t}$. Rather than performing fully Bayesian training, we adopt a post-hoc Laplace approximation to the posterior over network weights (Daxberger et al., 2021), which has recently been shown to provide accurate and computationally efficient uncertainty estimates for deep models.

Concretely, we first train a deterministic neural network by minimizing mean squared error on $\mathcal{D}_t$, which learns to predict docking scores from some given representation space $\phi(x) \in \mathbb{R}^d$. Let $\hat{\theta}_t$ denote the resulting parameters. We then approximate the posterior $p(\theta \mid \mathcal{D}_t)$ by a Gaussian centered at $\hat{\theta}_t$ with covariance given by the inverse Hessian of the negative log-likelihood,

$$p(\theta \mid \mathcal{D}_t) \approx \mathcal{N}(\hat{\theta}_t, H_t^{-1}),$$

where $H_t$ is estimated using a block-diagonal or Kronecker-factored approximation. This approximation yields a predictive distribution for each candidate molecule,

$$p(f(x) \mid \mathcal{D}_t) \approx \mathcal{N}(\mu_t(x), \sigma_t^2(x)),$$

which provides both predictive means and calibrated epistemic uncertainties.

Unless otherwise stated, the same surrogate architecture, training protocol, and Laplace approximation procedure are used across all experiments.

### A.4.2. ACQUISITION FUNCTION

Candidate selection is driven by the Upper Confidence Bound (UCB) acquisition function,

$$a_t(x) = \mu_t(x) + \beta_t \, \sigma_t(x),$$

where $\mu_t(x)$ and $\sigma_t(x)$ denote the predictive mean and standard deviation of the surrogate model, and $\beta_t > 0$ controls the exploration–exploitation tradeoff.

In standard Bayesian optimization, $a_t(x)$ is evaluated over the entire library. In BOBA, the same acquisition function is used, but inference is restricted to the subspace selected by the bandit layer. Thus, all methods share identical local decision rules; only the scope of inference and the global allocation of computation differ.

A.4.3. TRAINING AND UPDATE PROTOCOL

The surrogate model is retrained (or warm-started and updated) at each iteration using all available observations in $\mathcal{D}_t$.

**Model architecture.** The surrogate model is a fully connected feedforward neural network with two hidden layers of sizes 512 and 128, respectively, and ReLU nonlinearities. The network maps molecular feature vectors to a scalar prediction of docking score.

**Optimization.** All networks are trained using the Adam optimizer (Kingma & Ba, 2015) with learning rate $1 \times 10^{-3}$ and weight decay $1 \times 10^{-4}$. We employ a cosine annealing learning rate schedule (Loshchilov & Hutter, 2017) over the course of training. Each surrogate update consists of 500 training epochs with a batch size of 8192.

**Laplace approximation.** After deterministic training, we construct a post-hoc Bayesian neural network using a Laplace approximation over the final layer weights. We use a last-layer Laplace approximation (Daxberger et al., 2021) and refine the posterior for an additional 100 post-hoc optimization epochs. Predictive means and uncertainties are obtained from this approximate posterior and used to compute UCB acquisition values.

## A.5. Uncertainty Estimation Ablation

Because the acquisition function depends directly on predictive uncertainty, we evaluated whether the conclusions are specific to the Laplace approximation used in the main experiments. Figure 7 compares the Laplace approximation against SWAG (Maddox et al., 2019) and MC dropout (Gal & Ghahramani, 2016) for both full-library BO and BOBA across multiple libraries, targets, and values of $K$. Across these settings, the Laplace approximation provides the strongest or most consistent retrieval trajectories, particularly for full-library BO. The same qualitative trend is retained in BOBA, although performance varies more strongly across libraries and cluster granularities. These results support the use of the Laplace approximation as the default uncertainty estimator in the main benchmarks, while leaving more systematic uncertainty-calibration studies as a promising direction for future work.

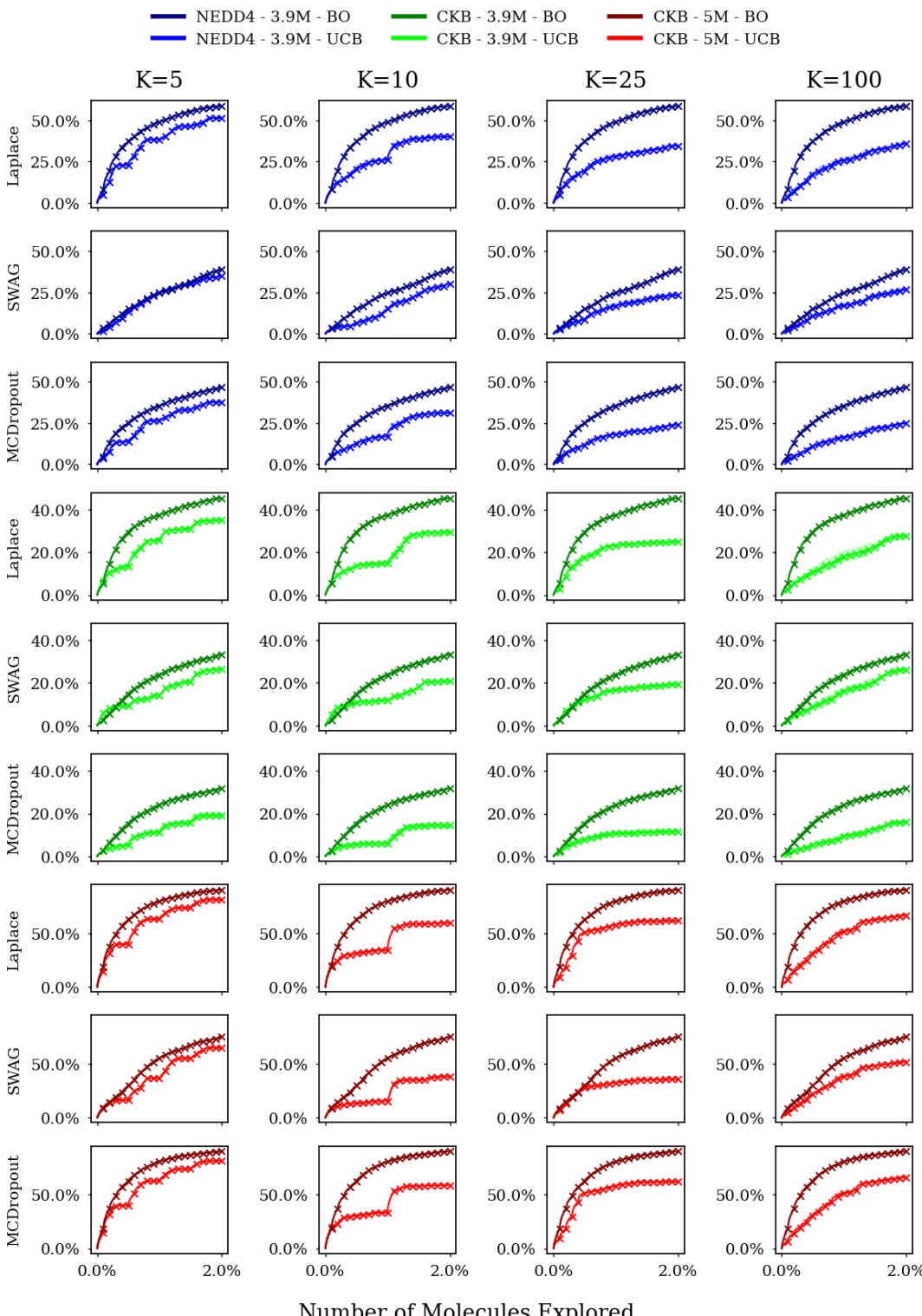

*Figure 7.* **Effect of uncertainty estimation on optimization performance.** Optimization trajectories for full-library BO and BOBA using Laplace approximation, SWAG, and MC dropout uncertainty estimates. Curves report retrieval of top-ranked candidates as a function of the number of molecules explored across ENAMINE-S-3.9M docked against NEDD4, ENAMINE-S-3.9M docked against CKB, and ENAMINE-5M docked against CKB. Columns correspond to different values of $K$.

## B. Physiochemical Descriptors

List of physiochemical properties used as heuristic features: aromatic proportion, aromatic ring count, atom count, bond count, chiral center count, doublebond stereoisomer count, electronegative atom count, formal charge, fraction of sp$^3$ carbon atoms, halogen atom count, hydrogen bond acceptor count, hydrogen bond donor count, heavy atom count, logD, logP, logS, molecular refractivity, molecular weight, negative charge count, nitrogen and oxygen atom count, positive charge count, Quantitative Estimate of Druglikeness (QED), ring count, rotatable bond count, sulfur atom count, Topological Polar Surface Area (TPSA). For the 5 million compounds, the properties were obtained from (Gorgulla et al., 2023). The quantitative properties were selected from all properties calculated in (Gorgulla et al., 2023).

# C. Additional Results

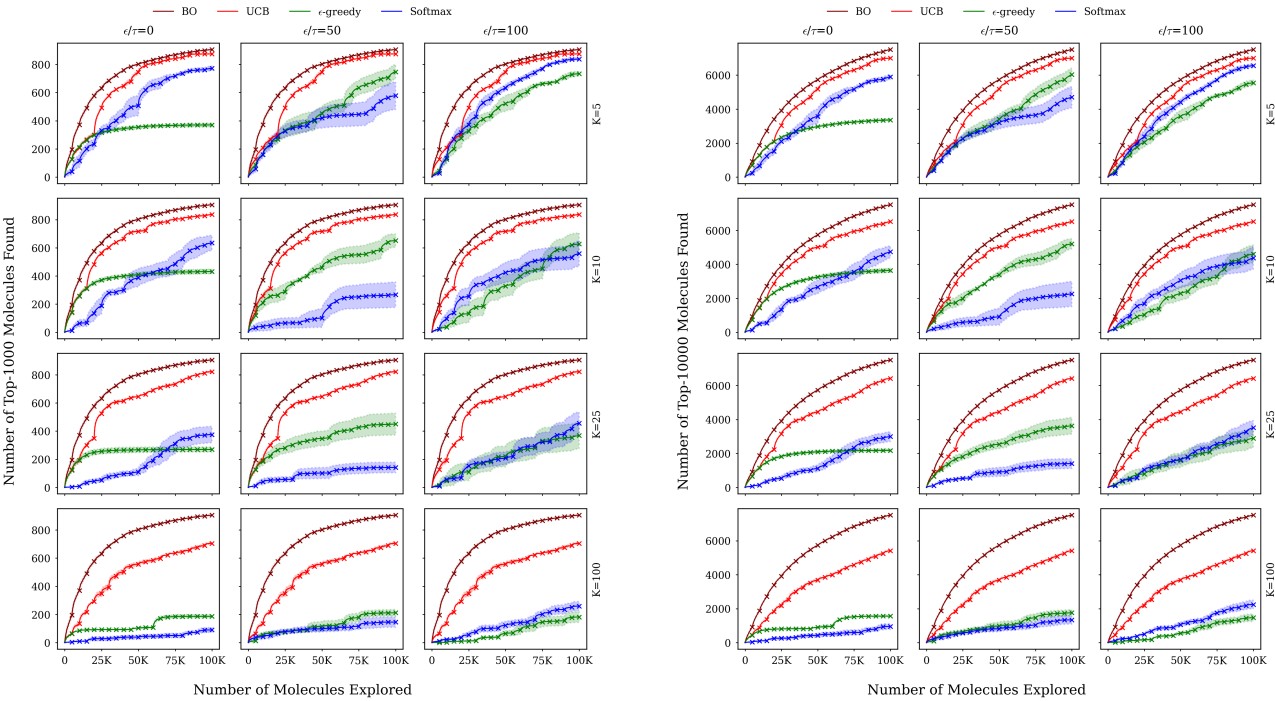

*Figure 8.* Trajectories of top-1000 and top-10000 molecules recovered for the ENAMINE-5M library (CKB).

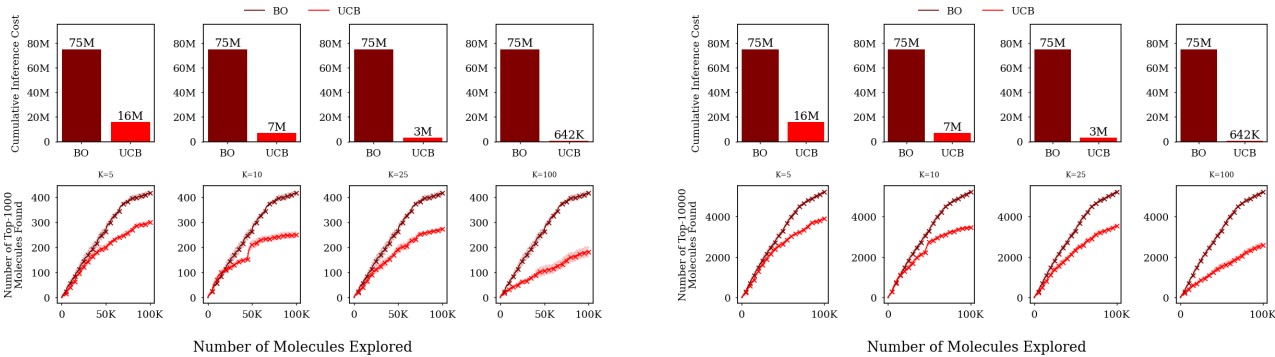

*Figure 9.* Trajectories of top-1000 and top-10000 molecules recovered for the ENAMINE-S-3.9M (CKB) library

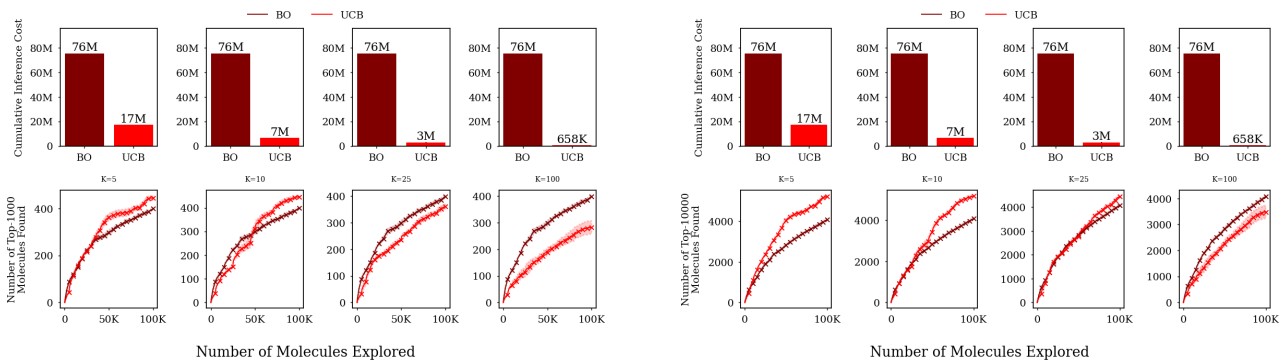

*Figure 10.* Trajectories of top-1000 and top-10000 molecules recovered for the ENAMINE-S-3.9M (NEDD4) library

## D. Arm Selection Frequencies

To assess whether UCB1 collapses onto a small number of initially high-reward partitions, we analyzed the fraction of selections assigned to each arm over the course of optimization. Figure 11 shows that UCB1 continues to distribute selections across multiple arms, even when the number of partitions is large. This behavior is consistent with the optimism bonus in UCB1, which keeps under-sampled partitions competitive until their empirical rewards are sufficiently well characterized. In the tested optimization budgets, we do not observe strong evidence of a winner's curse in which one early high-performing arm monopolizes the campaign after its best candidates are depleted.

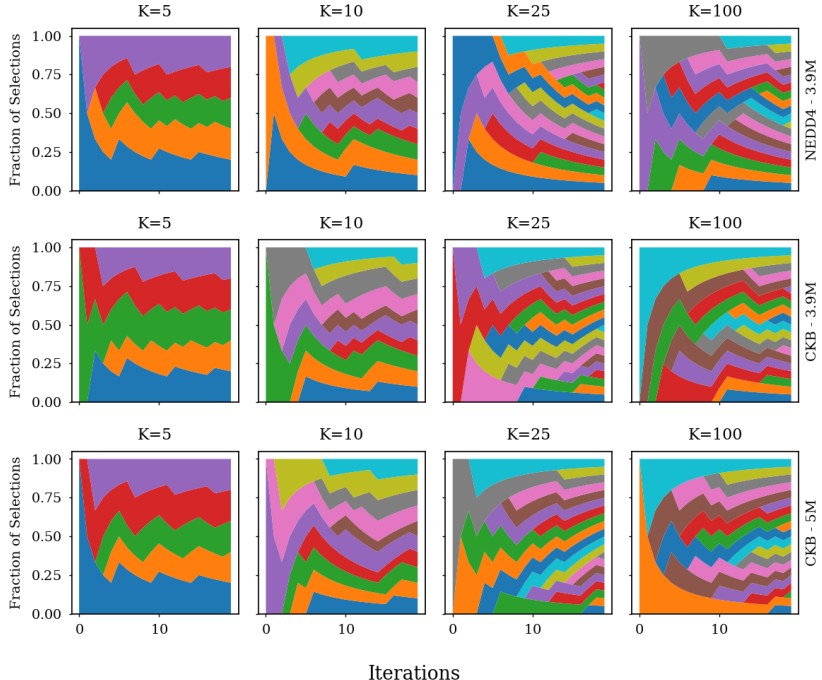

*Figure 11.* **Per-arm selection frequencies for BOBA with UCB1.** Each panel reports the fraction of selections assigned to each arm over optimization iterations, with arms color-coded by partition. Experiments are shown for ENAMINE-S-3.9M docked against NEDD4, ENAMINE-S-3.9M docked against CKB, and ENAMINE-5M docked against CKB, across $K \in \{5, 10, 25, 100\}$.

## E. Cost–Regret Tradeoff Analysis

This section provides a simplified theoretical analysis of the central cost–performance tradeoff in BOBA. The goal is not to fully model the molecular optimization problem, but to formalize the role of the number of clusters $K$ as a control parameter.

**Setup.** Consider a library of size $N$ partitioned into $K$ approximately equal clusters. At each of $T$ optimization rounds, full-library BO performs surrogate inference over all $N$ candidates, whereas BOBA performs inference over approximately $N/K$ candidates after selecting one arm. Thus, up to constants and batch-size factors shared across methods, the cumulative inference cost of BOBA scales as

$$C_{\text{inf}}(K) \asymp \frac{NT}{K}.$$

Increasing $K$ therefore reduces inference cost, but it also makes the bandit allocation problem harder because the algorithm must identify promising regions among more arms.

**Bandit term.** Under a standard stochastic bandit abstraction, the regret of UCB-type allocation over $K$ arms admits gap-dependent logarithmic bounds and gap-free bounds of order

$$R_{\text{bandit}}(K, T) = \tilde{\mathcal{O}}(\sqrt{KT}),$$

where logarithmic factors are suppressed. This term captures the loss from allocating rounds to suboptimal partitions before the bandit has confidently identified high-yield regions. The approximation ignores within-partition surrogate error and the mild non-stationarity induced by depletion of high-scoring candidates, but it isolates the effect of partition granularity.

**Joint objective.** Let $\lambda > 0$ denote the relative importance of inference cost compared with allocation regret. A simplified objective is

$$J(K) = \lambda \frac{NT}{K} + \sqrt{KT}.$$

The first term decreases with $K$, while the second increases with $K$.

Treating $K$ as continuous, we can minimize $J$ with respect to $K$ by differentiation:

$$\frac{dJ}{dK} = -\lambda NT K^{-2} + \frac{1}{2}\sqrt{T}K^{-1/2}.$$

Setting this derivative to zero yields

$$K^{3/2} = 2\lambda N \sqrt{T},$$

and therefore

$$K^* \asymp N^{2/3}T^{1/3},$$

up to constants and logarithmic factors.

**Interpretation.** This scaling law formalizes the empirical behavior observed in the main text. Larger libraries justify finer partitions because the inference savings from reducing the queried fraction of the library grow with $N$. At the same time, the optimal $K$ grows sublinearly, reflecting the increasing statistical burden of choosing among many arms. In practice, the constant hidden in the scaling depends on hardware, surrogate architecture, batch size, cluster balance, target difficulty, and the quality of the molecular representation. The analysis should therefore be interpreted as a design principle rather than a prescription for a single universal value of $K$.

# F. Distribution of Docking Scores

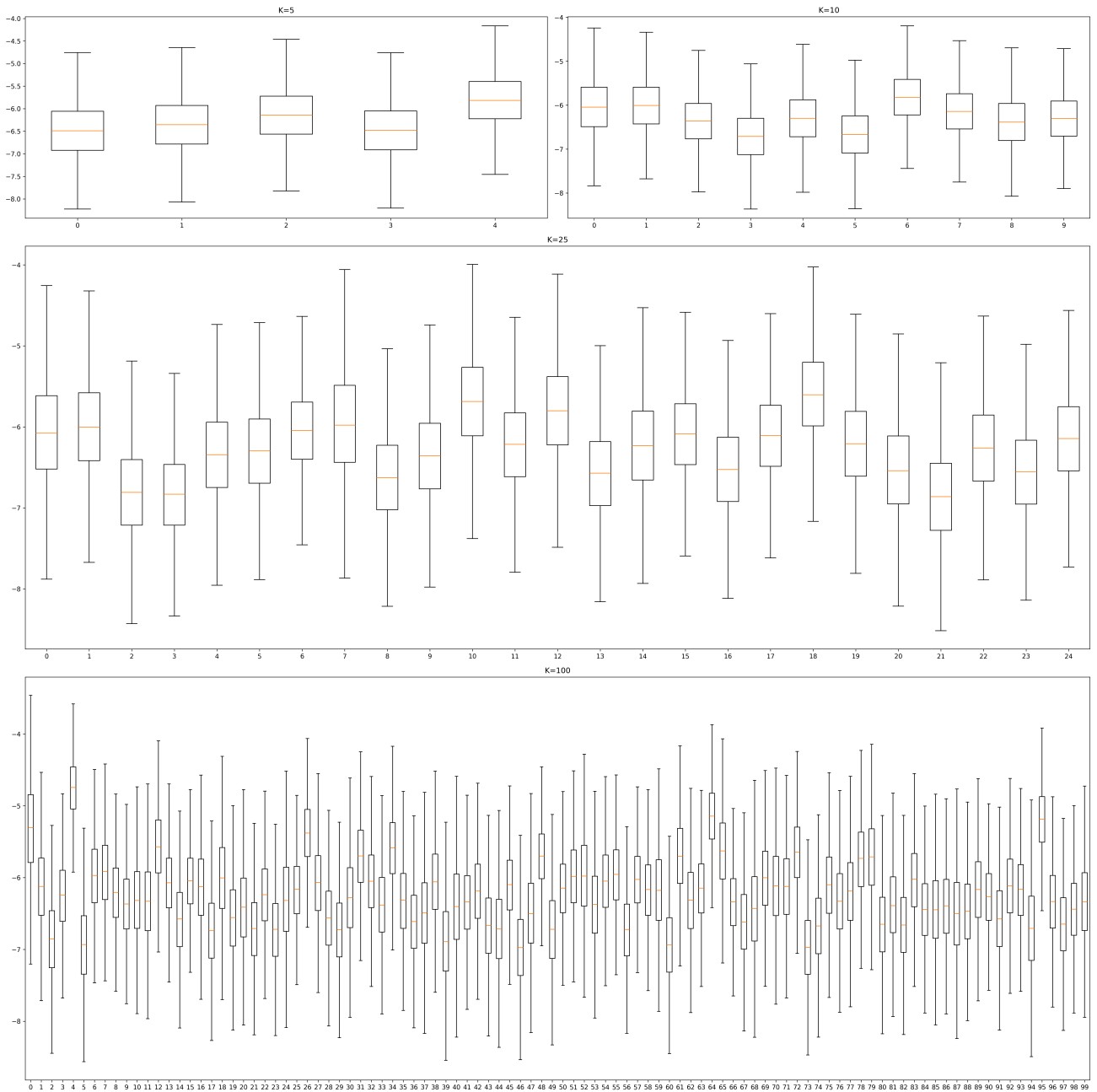

*Figure 12.* Distribution of docking scores for CKB-5M inside each cluster for different values of $K$. Outliers not shown.

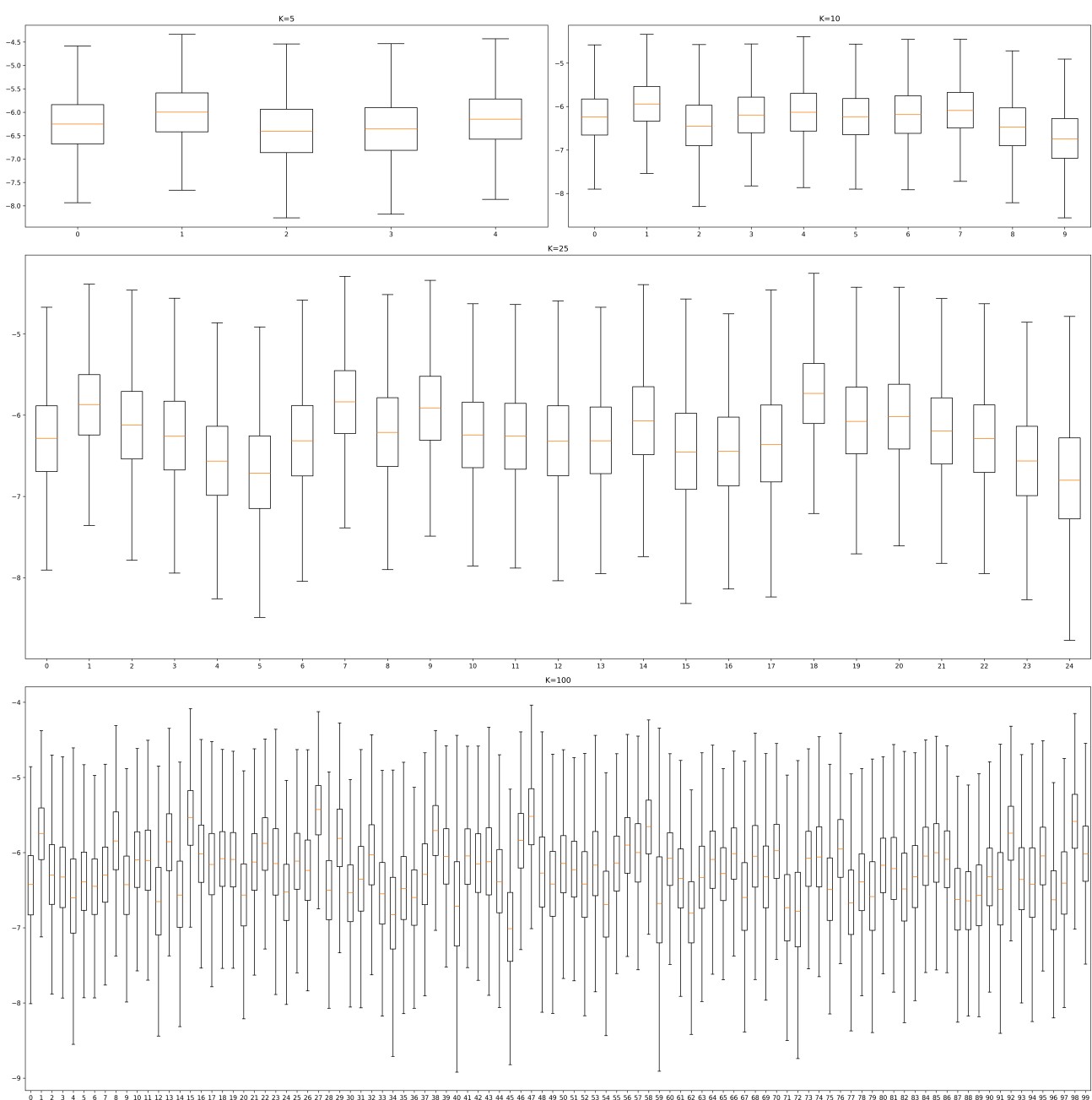

*Figure 13.* Distribution of docking scores for CKB-3.9M inside each cluster for different values of $K$. Outliers not shown.

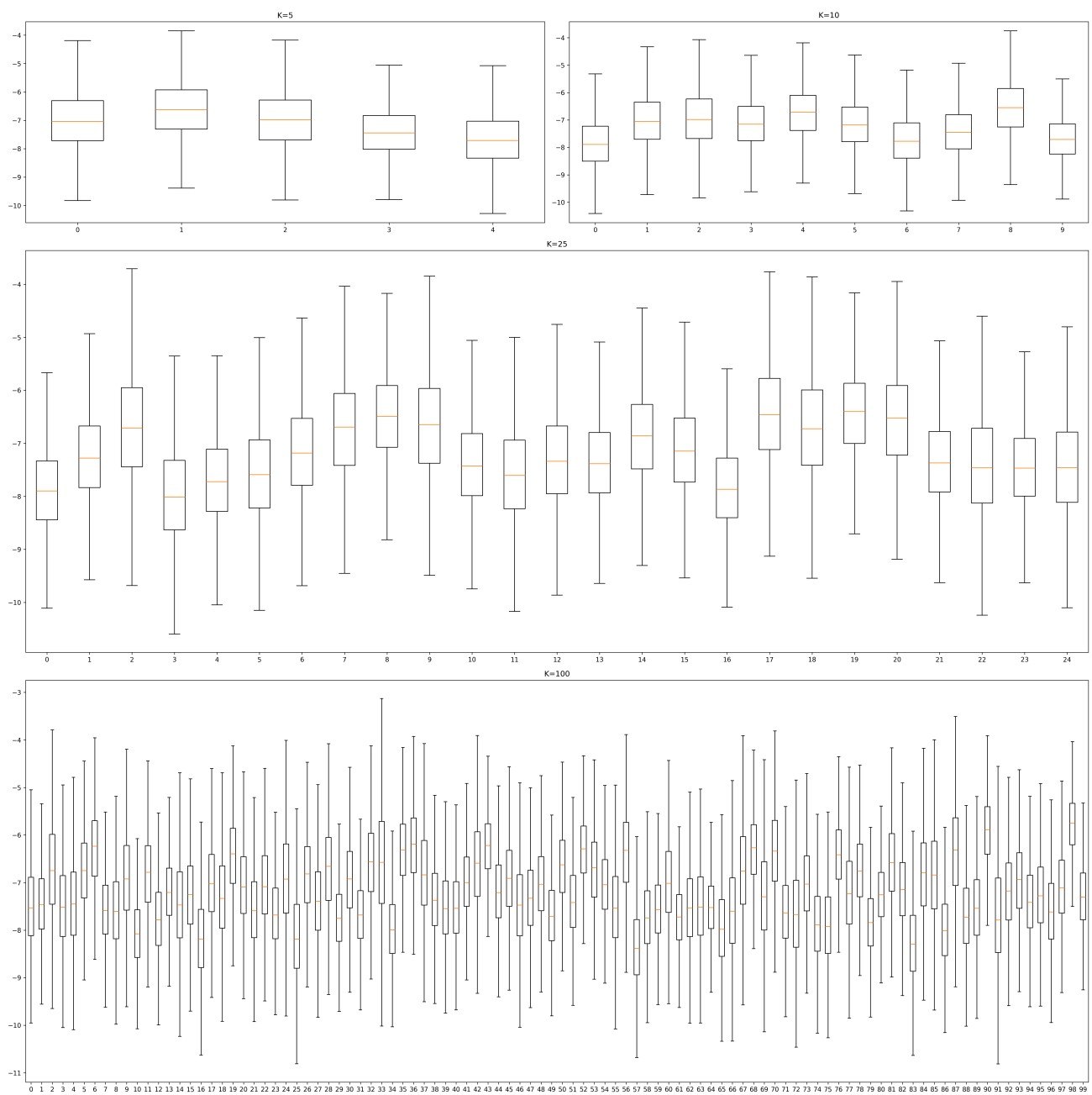

*Figure 14.* Distribution of docking scores for NEDD4-3.9M inside each cluster for different values of $K$. Outliers not shown.

## G. Physics-Based Docking

We randomly selected 5 million compounds from an enumerated library of 69 billion compounds from ENAMINE (Gorgulla et al., 2023). In addition, an independent set of 3.9 million compounds was randomly sampled from ENAMINE's S-class small-molecule database, which comprises approximately 3.9 billion unenumerated compounds, and retrieved in SMILES format. Only one stereoisomer or one tautomer was retained for each SMILES string to preserve database diversity. The compounds were then converted to three-dimensional structures and energy-minimized using RDKit (Landrum et al., 2023) and Open Babel (O'Boyle et al., 2011). The resulting library was subsequently docked against the target proteins with known active sites using the Uni-Dock small-molecule docking tool (Yu et al., 2023) on NVIDIA RTX 4500 Ada Generation GPUs.

Specifically, we docked the libraries against the proteins CKB and NEDD4. CKB is a cytosolic phosphotransferase that buffers cellular energy by catalyzing the reversible transfer of a phosphate group between ATP and creatine, thereby helping maintain ATP levels during fluctuating energy demand. (Bong et al., 2008) NEDD4 is a HECT-family E3 ubiquitin ligase that recognizes substrates via WW domains and catalyzes ubiquitin transfer to regulate protein stability, trafficking, and signaling. (Maspero et al., 2025)

We obtained docking scores against a Thymidylate Kinase (TMK) for the virtual library of 2M molecules from ENAMINE's HTS database (ENAMINE-HTS) from (Graff et al., 2021). TMK is a phosphotransferase that catalyzes the reversible transfer of a phosphate group from ATP to deoxythymidine monophosphate (dTMP), to form deoxythymidine diphosphate (dTDP) in the process of DNA synthesis. (Naik et al., 2015)

