# OpenReview forum: "Target-Aware Bandit Allocation for Scalable Surrogate Optimization in Chemical Space"
_ICML.cc/2026/Conference — ICML 2026 regular_

### Official Review · Reviewer_H4DA · 2026-03-05

**Soundness:** 3
**Presentation:** 3
**Significance:** 3
**Originality:** 3
**Overall Recommendation:** 5
**Confidence:** 3

**Summary:**

This paper introduces BOBa (Bayesian Optimization with Bandits), a framework designed to tackle a major computational bottleneck in virtual screening for ultra-large molecular libraries. As libraries grow to billions of compounds, just running a standard surrogate model across the entire dataset becomes prohibitively expensive. To resolve this, the proposed method partitions the chemical space into distinct clusters using K-means on molecular embeddings, treating each partition as an individual arm in a multi-armed bandit problem. Rather than scoring the entire library during every optimization round, BOBa employs standard bandit algorithms (such as UCB1, epsilon-greedy, and softmax) to selectively allocate inference and evaluation resources only to the most promising partitions. The key conceptual contribution is a dual-cost formulation that explicitly distinguishes the cost of the final objective evaluation (like expensive docking simulations) from the cost of surrogate model inference. The authors evaluate their approach on three protein targets across libraries of 2-5 million molecules sampled from a massive 69-billion compound collection. The results demonstrate that BOBa successfully recovers nearly the exact same top candidates as a full-library search while significantly reducing the inference cost (by a factor roughly proportional to the number of partitions).

**Compliance With Llm Reviewing Policy:**

Affirmed.

**Final Justification:**

I think this paper should be accepted for all of the reasons indicated in my initial review. I initially gave a score of 4, and have now updated my score to 5 to reflect that the author's rebuttal addressed all of the questions and concerns I had about the paper.

**Key Questions For Authors:**

**Q1:** Can you comment on extending BOBa to synthon-based or generative representations where the library is not explicitly enumerated? What are the main technical barriers? The Limitations section sketches this but I am interested in learning about more of the specifics.

**Q2:** The choice of K involves a clear performance-cost tradeoff (Figures 1-3). Do you have any specific insights or practical guidelines you could share for helping a practitioner in selecting K? e.g., given a specific evaluation budget T, library size N, and target hit count?

**Q3:** Figure 4 shows T5Chem embeddings outperform physicochemical descriptors for partitioning, but the gap is moderate. Since physicochemical descriptors are significantly cheaper to compute, when would you recommend each in practice?

**Q4:** Could the framework support adaptive partitioning that refines clusters during optimization based on observed rewards, rather than fixing them upfront?

**Limitations:**

yes

**Strengths And Weaknesses:**

### Soundness
The paper is technically solid. The dual-cost formulation in Section 3.1 is clean and well-motivated: as library sizes have exploded, the assumption that surrogate inference is cheap relative to evaluation no longer holds. I found the integration of Multi-Armed Bandits (MAB) with BO to be executed well. The experimental design is thorough, with three protein targets, multiple library sizes, several partitioning strategies, and proper ablations over $K$, feature spaces, and structured versus random partitions. Using actual docking scores as ground truth further strengthens this work. The Laplace approximation for BNN uncertainty is a standard and appropriate choice.

### Presentation
The writing is generally clear and well-structured. Section 3.1's distinction between evaluation and inference cost forms the conceptual core of the paper, and it is communicated well. Figure 1 effectively illustrates the performance-cost tradeoff.

### Significance
**Strengths:**
This paper addresses an important and timely problem. Make-on-demand chemistry has pushed molecular libraries into the billions, making the inference bottleneck a significant real-world issue. The practical payoff is clear from the experiments: BOBa achieves comparable hit recovery at a fraction of the inference cost, with the tradeoff easily tunable via the parameter $K$. Furthermore, the framework generalizes beyond drug discovery to any large discrete optimization problem where both evaluation and inference are costly.

**Weaknesses:**
The current restriction to strictly enumerable libraries limits near-term applicability to the absolute largest modern collections. The authors do acknowledge this in Section 6 and sketch how BOBa could extend to synthon-based representations, though that remains strictly a direction for future work.

### Originality
**Strengths:**
The dual-cost formulation is the primary novelty, and it provides a genuinely fresh angle motivated by practical realities that is missing from prior work on virtual screening / BO. Combining structure-aware partitioning with bandit allocation is a novel and creative approach. Additionally, the empirical finding that K-means clustering significantly outperforms random partitioning (Figure 3) is a valuable practical insight.

**Weaknesses:**
(Minor) The individual components are well-established (K-means, UCB1, BNNs with Laplace approximations), meaning the novelty lies almost entirely in the combination and the problem framing rather than in algorithmic invention. While the rubric notes this is an acceptable form of originality, it remains a minor weakness regarding methodological innovation.

---

> ### Author Rebuttal · Authors · 2026-03-31
>
> We thank the reviewer for the thoughtful questions regarding extensions, practical usage, and design choices.
>
> ---
> ## Extension to Synthon-Based Representations
>
> In this setting, 1) clustering would be performed over fragments and reactions, 2) a bandit algorithm would select a sub-library defined by combinations of fragment clusters, and 3) local optimization is performed within this sub-library.
>
> We anticipate several technical challenges in this extension:
> * The ***surrogate model would need to operate on fragment and reaction representations*** rather than full molecules. This likely requires learned pooling or factorization schemes that capture synthon-specific effects and pairwise or higher-order interactions.
> * Since ***clustering would occur at the fragment level***, the bandit would need to be hierarchical, and sequentially select fragment clusters.
> * ***Optimization within the selected sub-library*** could rely either on combinatorial enumeration of the sub-library, or on sampling-based approaches such as Tompson sampling over fragments.
>
> While this extension introduces additional complexity, we believe it is feasible and would retain the core benefit of BOBA: focusing computational resources on the most promising regions of an otherwise intractably large search space.
>
> ---
> ## Performance–Cost Tradeoff in Choosing the Number of Clusters
>
> We agree that BoBa introduces an inherent performance–cost tradeoff. By design, smaller values of K lead to improved optimization performance, approaching full-library BO in the limit *K*=1. As *K* increases, performance typically degrades.
>
> In practice, however, the ***choice of *K* is typically determined by budget constraints***. As the total evaluation budget *T* is limited by experimental and computational resources, *K* should be chosen as small as possible while keeping surrogate inference per iteration computationally feasible. In other words, *K* serves as a control parameter to balance optimization quality against computational cost. For a formal analysis of this tradeoff, please refer to our response to Reviewer P7my.
>
> ---
> ## Choice of Molecular Representation
>
> In our experiments, we have found that the choice of molecular representation plays two different roles in the two components of BoBa:
> * For ***clustering***, the choice of representation is of secondary importance. As shown in our benchmarks (see also response to Reviewer 2mEC), different reasonable representations lead to similar performance, provided they capture meaningful structure.
> * For ***surrogate modeling***, the choice is critical. We observe that foundation model embeddings consistently outperform simpler descriptors, in line with recent literature (in line with recent literature, e.g. https://doi.org/10.48550/arXiv.2402.05015).
>
> Given that high-quality embeddings are required for the surrogate model, we compute foundation model embeddings in all cases and recommend performing clustering in the same space as a practical and consistent choice. We have clarified this guideline in the revised manuscript.
>
> ---
> ## Adaptive Partitioning
>
> We agree with the reviewer that adaptive partitioning is conceptually interesting. However, ***BoBa intentionally uses fixed partitions*** as they can be reused across iterations and tasks, and avoids repeated clustering costs.
>
> Adaptive partitioning of a sparsely labeled library would require recomputing the partitioning at each iteration, which can be costly for large libraries. Moreover, it would introduce methodological challenges: The current bandit formulation assumes a fixed set of arms with accumulated statistics. Re-partitioning would effectively redefine the arms with imbalanced statistics, and likely, would lead to clusters without historical observations. This would require re-initialization and could result in a non-negligible fraction of the budget being spent on effectively random exploration. Alternatively, one may consider alternate bandit policies, e.g. based on the centroids and populations of each cluster. While feasible, this would introduce additional complexity and design choices.
>
> ---
>
> We thank the reviewer for the insightful questions. We believe the clarifications improve the manuscript and kindly ask for a reconsideration of the evaluation.

---

> > ### Author Rebuttal · Reviewer_H4DA · 2026-04-03
> >
> > The authors addressed each of my concerns and answered each of my questions from my review. I will update my score accordingly.

---

### Official Review · Reviewer_2mEC · 2026-03-11

**Soundness:** 3
**Presentation:** 3
**Significance:** 3
**Originality:** 3
**Overall Recommendation:** 5
**Confidence:** 3

**Summary:**

This paper studies a practical bottleneck in ultra-large virtual screening: when libraries become very large, full-library surrogate inference can itself be too expensive. To address this, the paper proposes BOBA, which partitions chemical space into persistent subspaces, uses a bandit to choose which partition to search, and then performs surrogate-based selection only within that partition. The experiments suggest that BOBA can retain much of the performance of full-library BO while reducing surrogate inference cost, and that both the bandit choice and the partition quality are critical.

**Compliance With Llm Reviewing Policy:**

Affirmed.

**Final Justification:**

The reviewer has addressed my concerns. Although my own confidence level remains moderate, I have decided to increase my score.

**Key Questions For Authors:**

1. Since the method depends strongly on partition quality, how sensitive is BOBA to the specific clustering algorithm and partition construction choices beyond the settings tested here?

2. Could the authors report end-to-end runtime or a simple breakdown of partitioning, surrogate retraining, and inference cost, in addition to inference counts?

3. The paper states that the current implementation assumes an explicitly enumerable library. Could the authors clarify more concretely what scale of library this preprocessing remains practical for in the current form?

**Limitations:**

yes

**Strengths And Weaknesses:**

Strengths

1. The paper addresses a clear and practically relevant problem setting, namely the regime where surrogate inference over the full candidate library is no longer negligible.

2. The method is simple and intuitive: it separates global allocation across regions of chemical space from local candidate selection within a region.

3. The empirical study is fairly solid. In particular, the ablations on bandit choice, structured vs. random partitions, and feature space help support the main claims.

Weaknesses

1. The method appears quite dependent on the quality of the initial partitioning. The results show that random partitions perform much worse, which suggests that the gains are sensitive to how the space is represented and clustered.

2. The cost analysis is mainly based on the number of surrogate inferences, rather than end-to-end runtime. This makes the practical speedup somewhat indirect.

3. The current evaluation is still restricted to settings where the full library can be explicitly enumerated, featurized, and clustered. The paper acknowledges this limitation directly.

---

> ### Author Rebuttal · Authors · 2026-03-31
>
> We thank the reviewer for the constructive feedback, particularly regarding partition quality, runtime evaluation, and scalability.
>
> ---
>
> ## Sensitivity to Partition Quality
>
> To evaluate this sensitivity empirically, we ***have conducted additional ablations*** comparing:
> * *K*-means clustering using cosine similarity as an alternate distance metric
> * *K*-means clustering on physicochemical descriptors
> * random partitioning
>
> The results are shown in [Fig. R4](https://anonymous.4open.science/r/boba-rebuttal/Fig_R4.pdf).
>
> Overall, our findings confirm the initial observation that structured clustering consistently outperforms random partitioning. Across ***reasonable representations and clustering methods, performance differences are moderate***. This suggests that, while partition quality matters, BoBa is not overly sensitive to the exact choice of the clustering methods, provided that meaningful structure is captured.
>
> We have added this conclusion to the main text of the manuscript, and have added the full analysis to the Appendix.
>
> ---
>
> ## Runtime Analysis
>
> For a detailed analysis of runtime as a function of library size, including breakdown into embedding calculation, clustering, surrogate training, and inference, see our response to reviewer P7my.
>
> ---
>
> ## Scaling k-Means Clustering to Large Library Sizes
>
> In its current form, the k-means clustering step requires explicit enumeration of the full library. At present, explicit enumeration has been demonstrated for libraries of e.g. ~100B molecules. At this scale, applying the BoBa preprocessing (featurization and K-means clustering) requires careful computational design, but remains feasible in practice:
> * Embedding computation can be parallelized.
> * Storage requirements for embeddings are on the order of ~300 TB (float32, uncompressed)
> * Clustering via *K*-means scales linearly in the number of data points (O(*n*)), and can be efficiently implemented using distributed or streaming variants.
>
> Based on these considerations, we argue that whenever one can afford explicit library enumeration, BoBa preprocessing should be technically feasible, although potentially resource-intensive.
>
> A key design ***choice of BoBa is that this preprocessing step is performed once per library***. The resulting clustering can then be re-used across multiple optimization campaigns, e.g. docking against different drug targets. This amortization substantially improves practical utility.
>
> For larger libraries where explicit enumeration and clustering are unfeasible, BoBa can be extended to a fragment-based representation which avoids explicit enumeration. We outline such an extension in our response to reviewer H4DA.
>
> ---
>
> We thank the reviewer again for the helpful suggestions and hope the additional analysis addresses the concerns. We kindly ask for reconsideration of the score.

---

> > ### Author Rebuttal · Reviewer_2mEC · 2026-04-02
> >
> > My initial score is already quite high, so I’ve decided to keep my score.

---

> > > ### Author Response · Authors · 2026-04-07
> > >
> > > We thank the reviewer for confirming that our additional experiments and clarifications have adequately addressed their concerns.
> > >
> > > Given that option (a) was selected, which explicitly states: 'please consider adjusting your score accordingly', we would appreciate if the reviewer could either articulate any remaining concerns, or reconsider their score.

---

### Official Review · Reviewer_P7my · 2026-03-12

**Soundness:** 3
**Presentation:** 3
**Significance:** 2
**Originality:** 2
**Overall Recommendation:** 4
**Confidence:** 4

**Summary:**

This paper addresses the computational challenge of surrogate-based optimization over ultra-large
  molecular libraries (billions to trillions of compounds) for virtual screening in drug discovery. The
  authors observe that at such scales, not only are black-box evaluations (e.g., molecular docking)
  expensive, but even a single forward pass of a surrogate model over the full library becomes a
  computational bottleneck, violating a core assumption of standard active learning and Bayesian optimization
   pipelines. To address this dual-budget constraint, the authors propose BOBA (Bayesian Optimization with
  BAndits), a hierarchical framework that partitions the molecular library into K chemically coherent
  subspaces (via clustering in a molecular feature space), treats each subspace as an arm in a multi-armed
  bandit problem, and uses bandit-based allocation to decide which subspace to perform surrogate inference
  and candidate selection within at each iteration. Experiments on real-world Enamine libraries (2M–5M
  compounds) docked against multiple protein targets systematically investigate the effects of bandit
  algorithm choice, partition granularity, partitioning strategy (structured vs. random), and feature space
  (foundation model embeddings vs. physicochemical descriptors). The key findings are that UCB1-based
  allocation with structured partitions in T5Chem embedding space retains near-full-library BO performance
  at substantially reduced inference cost, and that a tunable, sublinear tradeoff between optimization
  performance and inference cost exists.

**Compliance With Llm Reviewing Policy:**

Affirmed.

**Final Justification:**

The authors have provided substantial additional work addressing all major concerns. The 100M-scale experiments (with 100k/1M/10M sub-libraries) convincingly demonstrate scalability beyond the original 2-5M setting. The formal cost-performance tradeoff analysis with the derived scaling law for optimal K meaningfully elevates the contribution beyond a purely empirical one. The per-arm selection frequency analysis adequately addresses the non-stationarity concern within the current experimental regime, and the explanation for why fragment-based baselines are not directly comparable is reasonable.

**Key Questions For Authors:**

1. The paper claims BOBA is designed for billion- to trillion-scale libraries, but experiments are
  conducted only at the 2M–5M scale. Can the authors provide evidence that: (a) full-library surrogate
  inference is already a genuine computational bottleneck at 5M scale (e.g., wall-clock time comparisons)?
  (b) K-means clustering over foundation model embeddings is computationally feasible at billion scale? If
  the method cannot be validated at its claimed target scale, this appears to undermine the core motivation.
   Clarification on these points could change my assessment of the paper's significance.

  2. Does the current reward definition (batch-average docking score) lead to a "winner's curse"
  problem—where the bandit persistently selects an initially high-performing subspace whose high-quality
  molecules are gradually depleted? Can the authors provide an analysis of per-arm selection frequencies
  over the course of optimization to demonstrate that UCB1 achieves reasonable exploration in practice? This
   would help clarify whether the non-stationarity concern is practically relevant within the current
  experimental setup.

  3. In 4.3, T5Chem embeddings are used for clustering. What representation does the surrogate model use
  for training and inference? If the same representation is used for both clustering and surrogate modeling,
   is there a risk of circular dependency? If different representations are used, how does the misalignment
  between the clustering space and the surrogate's feature space affect performance? A clear answer would
  help assess the generality of the feature space findings.

  4. The experiments fix T=20 rounds with B=5,000. For K=100, UCB1 requires pulling each arm once for
  initialization, meaning over half of the 20 rounds are devoted to initialization rather than adaptive
  selection. How sensitive are the relative performances of different bandit algorithms to the T/K ratio?
  Experiments or discussion on larger T or varying T/K ratios would help assess the robustness of the
  conclusions. This is particularly important for evaluating whether UCB1's advantage would persist or
  diminish under different budget regimes.

**Limitations:**

yes

**Strengths And Weaknesses:**

Strengths：
1. Well-motivated and clearly formalized problem setting. The paper precisely identifies a real and
  increasingly pressing bottleneck in ultra-large library virtual screening, the computational cost of
  surrogate inference itself. The formalization of this "dual-budget constraint" (evaluation budget +
  inference budget) is a meaningful extension of standard BO/active learning frameworks that directly
  responds to the practical challenges arising as compound libraries scale from billions to trillions.

  2. Systematic and well-structured experimental design. The experiments are organized in a logically
  progressive manner, each addressing a distinct and important question: (1) the impact of bandit algorithm
  choice (4.1); (2) persistent structured partitions vs. dynamic random subsampling (4.2.1); (3)
  structured clustering vs. random assignment (4.2.2); (4) the effect of feature space (4.3); and (5)
  robustness across targets and difficulty regimes (4.4). Each experiment includes appropriate controls and
   baselines, and the ablation study is thorough.

  3. Clean, modular framework design. The core idea of decoupling global resource allocation (bandit) from
  local optimization (surrogate-based BO) is conceptually elegant and easy to understand. The modular
  design, where the partitioning method, bandit algorithm, surrogate model, and acquisition function are all
  interchangeable, provides a flexible experimental platform for future research.

  4. Benchmarking on real-world drug discovery data. The experiments are conducted on real Enamine compound
   libraries with ground-truth docking scores obtained via physics-based docking (Uni-Dock) against multiple
   protein targets (CKB, NEDD4, TMPK), rather than synthetic benchmarks. This enhances the practical
  relevance of the findings.

Weaknesses
  1. Significant gap between experimental scale and claimed application scope. The paper repeatedly
  emphasizes that BOBA is designed for billion- to trillion-scale libraries, yet all experiments are
  conducted on libraries of only 2M–5M compounds. At this scale, it is questionable whether full-library
  surrogate inference constitutes a genuine computational bottleneck. The paper reports only the number of
  surrogate inferences as a proxy for cost, without providing wall-clock time comparisons or actual
  computational resource measurements. This makes it difficult to assess the practical significance of the
  inference cost savings. Furthermore, the feasibility of the clustering step itself (K-means over
  foundation model embeddings) at trillion scale is not addressed.

  2. Reward non-stationarity is acknowledged but not addressed. BOBA uses the batch-average docking score
  as the arm reward. As optimization progresses, high-quality molecules within a subspace are gradually
  depleted, causing the reward distribution to shift over time (rotting rewards). While the authors mention
  rotting bandits as future work in the conclusion, the current framework does not account for this at all.
  Although this may have limited impact within the T=20 experimental setting, it becomes increasingly
  problematic at the larger scales the paper targets.

  3. Missing comparisons with most relevant baselines. The Related Work section discusses several
  bandit-based and active learning methods that operate in synthon space for ultra-large library screening
  (Klarich et al., 2024; Zhao et al., 2025; Grigg et al., 2025), but none of these are included as
  experimental baselines. While the authors note that these methods operate over different structural
  decompositions of the library, they address the same application problem (efficient virtual screening of
  ultra-large libraries), and direct comparison would substantially strengthen the paper's empirical
  contribution.

  4. Purely empirical contribution with no theoretical analysis. The paper's contributions are entirely
  empirical. The claimed "sublinear decay" of optimization performance with increasing K is supported only
  by visual inspection of plots, with no formal analysis or bounds. Given the rich theoretical toolkits
  available for both bandit algorithms and Bayesian optimization, even an analysis under simplifying
  assumptions would significantly strengthen the contribution.

---

> ### Author Rebuttal · Authors · 2026-03-31
>
> We thank the reviewer for the detailed and insightful feedback. The comments regarding scalability, non-stationarity, baselines, and theoretical grounding are particularly valuable. We address the individual points of criticism below.
>
> ---
>
> ## Scalability and Cost
> To evaluate the scaling of BoBa to large libraries, ***we include a new set of experiments on a 100M-sized candidate library***, along with sub-libraries of 10M, 1M and 100k. Optimization behavior and detailed runtime measurements are shown in [Fig. R2](https://anonymous.4open.science/r/boba-rebuttal/Fig_R2.pdf) and [Tab. R2](https://anonymous.4open.science/r/boba-rebuttal/Tab_R2.pdf).
>
> Key Findings:
> * As library sizes increase, BoBa’s ability to approximate the behavior of full-library BO does not deteriorate. For any library size, we observe the tradeoff between computational cost and performance. We have discussed this behavior in the paper.
> * The overall run time is dominated by embedding computation, which is a one-time cost, and can be re-used across optimization tasks on the same library. Within the optimization loop, run times are dominated by surrogate model inference.
>
> Regarding the scalability of the k-means clustering step, please refer to our response to reviewer 2mEC.
>
> ---
>
> ## Non-Stationarity of the Reward
>
> As discussed in the manuscript, the bandit reward is theoretically non-stationary due to depletion effects. In our practical regime, however, two aspects limit this effect.
> * The total evaluation budget T is small compared to the library size due to the cost of evaluations.
> * The bandit operates in an exploratory setting.
>
> To confirm the second aspect, we have ***empirically analyzed per-arm selection frequencies***. The results are shown in [Fig. R3](https://anonymous.4open.science/r/boba-rebuttal/Fig_R3.pdf).
>
> These results indicate that the ***UCB1 algorithm maintains sufficient exploration***, and we do not observe strong evidence of a “winner’s curse” in our experiments. We have added this analysis to the revised manuscript.
>
> ---
>
> ## Missing Comparisons to Fragment-Based Optimization
>
> We agree that fragment-based approaches that rely on Thompson sampling (e.g. the methods proposed by Klarich et al., Grigg et al., Zhao et al.) are an important baseline. However, these methods operate on combinatorial fragment libraries, whereas the datasets used herein are non-enumerated collections or subsamples of enumerated collections. Therefore, ***direct comparison is unfortunately unfeasible***. We identify this as an important direction for future work.
>
> ---
>
> ## Theoretical Guarantees
>
> We agree that a theoretical treatment would strengthen the contribution of BoBa. In response, we have ***derived a formal analysis of the cost–performance tradeoff in BoBa*** under simplifying assumptions. We provide the full derivation in [Analysis 1](https://anonymous.4open.science/r/boba-rebuttal/Analysis_1.pdf).
>
> A central result of this analysis is that the ***number of clusters K acts as a control parameter*** governing the trade-off between cost and performance. Specifically, when introducing a weight to balance inference cost and bandit regret, the optimal number of clusters scales as
>
> $K^\ast \asymp N^{2/3} T^{1/3}$
>
> where $N$ is the library size and $T$ is the evaluation budget. This ***heuristic scaling law*** formalizes the intuition behind BoBa. Increasing $K$ reduces inference cost but makes partition selection more challenging. We included this analysis in the revised manuscript.
>
> ---
>
> ## Embedding Choice for Clustering and Surrogate Modeling
>
> We confirm that, in all experiments, ***clustering and surrogate modeling use the same T5Chem embedding space***. We clarified this in the paper.
>
> Yet, the reviewer raises an important point: Both clustering and surrogate modeling rely on the assumption that similarity in embedding space correlates with similarity in objective values. However, they operate on different notions of similarity: k-means clustering captures global structure of the embeddings, while the surrogate model can exploit local variations. That said, the main risk in using the same embedding is that biases may propagate through both steps.
>
> To probe this, we conducted ***an additional ablation study*** (see response to reviewer 2mEC). These data suggest that, while partition quality matters, BoBa is not highly sensitive to the exact nature of the clusters. We attribute this robustness to the combination of an expressive global surrogate model and the explorative bandit strategy. We expanded this discussion in the manuscript.
>
> ---
>
> ## Bandit Initialization
>
> In our work, we adopt a ***single-round initialization strategy***, where each arm is seeded with B/K samples from the resp. cluster. This reduces initialization overhead. We have updated the paper to clarify this.
>
> ---
>
> Overall, we believe the additional experiments and clarifications strengthen the paper. We kindly ask the reviewer to reconsider their score.

---

> > ### Author Rebuttal · Reviewer_P7my · 2026-04-05
> >
> > The authors have provided substantial additional work addressing all major concerns.  I raise my  score to 4.

---

### Official Review · Reviewer_SzzZ · 2026-03-13

**Soundness:** 2
**Presentation:** 3
**Significance:** 3
**Originality:** 2
**Overall Recommendation:** 2
**Confidence:** 4

**Summary:**

This paper introduces a bandit-guided surrogate optimization method that adaptively allocates computation across partitions of the action space to reduce inference cost. The experiments on optimization in the chemical space demonstrate a tunable trade-off between optimization performance and inference cost, shaped by both the bandit exploration strategy and the structure of the partitioning scheme.

**Compliance With Llm Reviewing Policy:**

Affirmed.

**Key Questions For Authors:**

Please refer to the strengths and weaknesses above. In addition, accurately quantifying surrogate model uncertainty is crucial for the performance of the acquisition function. Therefore, it can be very helpful for the paper to not only consider the Laplace approximation, but also other reasonable alternatives such as MC dropout, deep ensembles, and SWAG.

**Limitations:**

yes

**Strengths And Weaknesses:**

Strengths: The paper is clearly written, the motivation is well articulated and the problem is important: scalable inference is needed when the candidate set is large.

Weaknesses: Novelty and soundness: The paper spends about two pages on preliminaries and problem setup, but less than one page on the proposed method. As presented, the main novelty seems to be an algorithm that uses existing bandit methods to identify a promising cluster or local region, and then performs Bayesian optimization within that region. However, this kind of hierarchical optimization strategy has already been studied in the scalable BO literature. For example, [1, 2] first identify a region of interest and then run BO within that region. In your setting, the chemical compounds could similarly be mapped into a continuous latent space, after which these existing methods could be applied directly. For this reason, I do not yet find the novelty sufficiently convincing. I am also concerned about the experimental section. Most of the experiments appear to be ablations, rather than comparisons against strong existing scalable BO baselines. As a result, the empirical evidence is not strong enough to support the claimed contribution.


[1] Eriksson, David, et al. "Scalable global optimization via local Bayesian optimization." Advances in neural information processing systems 32 (2019).
[2] Li, Wenxuan, Taiyi Wang, and Eiko Yoneki. "Navigating in High-Dimensional Search Space: A Hierarchical Bayesian Optimization Approach." arXiv preprint arXiv:2410.23148 (2024).

---

> ### Author Rebuttal · Authors · 2026-03-31
>
> We thank the reviewer for their careful reading of our manuscript, and for raising important conceptual similarities to previous works. We appreciate the reviewer’s constructive comments and hope that the clarifications below address these concerns and justify a higher assessment of our work.
>
> ---
>
> ## Novelty and Relation to Scalable Bayesian Optimization
>
> We agree that BoBa shares a high-level similarity with hierarchical optimization strategies such as TurBO and HiBO, in that both approaches identify regions of interest and perform local optimization. However, the core distinction lies in the ***problem regime and the primary challenge***.
>
> Our setting focuses on large discrete candidate libraries, where we train a global surrogate model on a shared molecular representation space, and the dominant ***bottleneck is evaluating this global surrogate across the full candidate set***. BoBa addresses this bottleneck by maintaining the global surrogate model, and using a Bandit strategy to restrict inference to a dynamically selected subset of the discrete space.
>
> In contrast, methods like TurBO and HiBO are motivated by continuous, high-dimensional optimization, where ***fitting such a global surrogate model is the primary challenge***. Therefore, these methods identify regions of interest and construct local surrogate models.
>
> This distinction is central to our contribution, and has now been clarified in the revised manuscript.
>
> ---
> ## Comparison to Latent-Space Optimization Approaches
>
> We respectfully note that, in the given discrete optimization setting, applying BO in a continuous latent space requires the subsequent projection of the optimum back to the discrete candidate space. This non-bijective decoding step introduces an optimization–evaluation mismatch, which has been identified as a major source of of suboptimal optimization behavior in the literature (e.g. https://doi.org/10.1016/j.neucom.2019.11.004, https://arxiv.org/abs/2310.20258v3). As a result, ***methods like TurBO and HiBO cannot be directly applied to our setting without incurring this mismatch***. In contrast, discrete optimization approaches like BoBa avoid this issue by operating directly on the discrete compound library.
>
> To further strengthen the empirical evidence, we have ***extended our experiments to a broader range of virtual libraries*** (discrete candidate sets in feature space) ***and protein targets*** (objective functions over this feature space). Overall, we observe strong performance of BoBa relative to full-library BO, with the latter showing notable variability across libraries. This supports our original hypothesis that the diversity and distribution of discrete candidates over feature space critically influence optimization performance, which further motivates discrete optimization approaches. The full results are shown in [Tab. R1](https://anonymous.4open.science/r/boba-rebuttal/Tab_R1.pdf), and have been incorporated into the revised manuscript.
>
> ---
> ## Importance of Uncertainty Quantification
>
> We thank the reviewer for highlighting the importance of uncertainty estimation. We have ***extended our experiments to empirically evaluate the influence of uncertainty quantification***. Here, we compare the optimization behavior when using the Laplace approximation with MC Dropout and SWAG. Across multiple compound libraries, we observe that:
> * In full-library BO, the Laplace Approximation consistently provides higher sample efficiency than MC dropout or SWAG.
> * This trend largely translates to BoBa, although the cross-library variance is larger.
>
> Full results are shown in [Fig. R1](https://anonymous.4open.science/r/boba-rebuttal/Fig_R1.pdf).
>
> We included this discussion in the revised version of the manuscript, and included all experimental results in the Appendix.
>
> ---
>
> In summary, we believe the clarified distinction in problem setting, together with the expanded experimental analysis, strengthens the contribution. We kindly ask the reviewer to reconsider their assessment in light of these clarifications.

---

> > ### Author Rebuttal · Reviewer_SzzZ · 2026-04-04
> >
> > Thank you for your detailed response and for carefully addressing my comments. I appreciate the additional uncertainty quantification experiments. That said, my main concern about novelty and baseline comparison still remains. Given the authors’ acknowledgment that BoBa bears high-level similarity to hierarchical optimization approaches such as TurBO and HiBO, I believe a direct empirical comparison with these methods is necessary. Mapping discrete compound representations into a continuous embedding space is already a common practice in drug discovery, regardless of potential debates about its effectiveness. Therefore, evaluating established BO methods such as TurBO and HiBO in the continuous embedding space would provide a more convincing assessment of the proposed method’s advantages, rather than relying on the claim that no existing baseline is applicable. Therefore I am leaning to keep my rating.

---

> > > ### Author Response · Authors · 2026-04-07
> > >
> > > We thank the reviewer for the thoughtful suggestion to compare our approach with existing hierarchical optimization methods. In response, we carried out a ***systematic empirical evaluation of the dynamic trust-region strategy used in TuRBO*** within our discrete optimization setting. Specifically, we adapted TuRBO to operate in the same continuous embedding space, namely the T5Chem latent space, while keeping the surrogate model, acquisition function, batch size, and other experimental settings fixed to ensure a direct and controlled comparison.
> > >
> > > ---
> > >
> > > **Table R3: Optimization performance of BoBa and TuRBO on a random subset of 100K from the Zinc library docked against AmpC.**
> > > We report the area under the optimization curve for top-M candidate retrieval and the number of retrieved candidates under a fixed evaluation budget of 20 evaluations. Surrogate model, acquisition function, batch size, and all other experimental settings were kept constant across methods. TR = Trust Region.
> > >
> > > | Method                  | Top-100 Retrieval AUC | Number of Top-100 Candidates | Top-1000 Retrieval AUC | Number of Top-1000 Candidates |
> > > |------------------------|----------------------|-----------------------------|------------------------|-------------------------------|
> > > | TuRBO (TR Size 500)    | 0.0                  | 0.0                         | 0.000158               | 0.4                           |
> > > | TuRBO (TR Size 1000)   | 0.0                  | 0.0                         | 0.000209               | 0.6                           |
> > > | TuRBO (TR Size 2000)   | 0.0                  | 0.0                         | 0.000051               | 0.2                           |
> > > | BoBa (k=5)             | 0.33                 | 57.0                        | 0.297                  | 547.0                         |
> > > | BoBa (k=100)           | 0.11                 | 19.4                        | 0.092                  | 182.2                         |
> > >
> > > ---
> > >
> > > The results (Table R3) show that, across a range of trust region scales, under these settings, TuRBO remains unable to recover more than a very small fraction of the top-1000 candidates from the virtual library. This likely reflects the difficulty of applying a locally expanding trust-region strategy to a large chemical search space whose objective landscape is highly irregular and only imperfectly captured by local continuity in the embedding space. In that sense, the observed behavior is not necessarily a general limitation of TuRBO, but rather a consequence of the mismatch between its local-search assumptions and the effectively discrete structure of virtual library optimization. By contrast, BoBa’s multi-armed bandit formulation explicitly allocates evaluations across multiple regions, enabling more systematic global exploration while still refining promising areas, which leads to improved coverage of high-scoring candidates
> > >
> > > ---
> > >
> > > We thank the reviewer again for prompting this comparison, which we feel has sharpened the manuscript by clarifying the distinction between local trust-region optimization and bandit-based exploration in this problem setting. The added results provide a direct empirical basis for this distinction under matched experimental conditions, and they help explain why broader exploration is especially important in large-scale virtual library search. We hope this additional analysis addresses the reviewer’s concern and supports reconsideration of the score.

---

### Decision · Program_Chairs · 2026-04-30

**Decision:**

Accept (regular)

**Comment:**

This paper proposes BOBA, a bandit-based framework for computationally efficient surrogate optimization framework. Not only the observation of true objective, but also the simulation is computationally bottleneck in large-scale chemical optimization and the paper addressed it. The framework splits the large simulation space into $K$ subspaces and adaptively choose subspaces to build a global surrogate model in a computational efficient manner.

Three among the four reviewers are convinced and pushed the paper, while the other one reviewer is not. Reviewer SzzZ considers the additional comparison with baseline shown during author response is unconvincing given the baseline requires careful hyperparameter tuning. That said, the tone of reviews is generally positive. Reviewer P7my considers the derivation of motivation while these are addressed during the author response. The other reviewers also described limitations and generally satisfied during the response period. Given these, I think the paper tackles a well-motivated questions and will recommend an acceptance.